# A systematic review and embryological perspective of pluripotent stem cell-derived autonomic postganglionic neuron differentiation for human disease modeling

Thomas A Bos[1]*, Elizaveta Polyakova[1], Janine Maria van Gils[1], Antoine AF de Vries[2], Marie-José Goumans[3], Christian Freund[1,4], Marco C DeRuiter[1,5], Monique RM Jongbloed[1,2,5]*

[1]Department of Anatomy and Embryology, Leiden University Medical Centre, Leiden, Netherlands; [2]Department of Cardiology, Leiden University Medical Centre, Leiden, Netherlands; [3]Department of Cell and Chemical Biology, Leiden University Medical Centre, Leiden, Netherlands; [4]Leiden hiPSC Centre, Leiden University Medical Centre, Leiden, Netherlands; [5]Centre for Congenital Heart Disease Amsterdam-Leiden (CAHAL), Leiden, Netherlands

*For correspondence:
t.a.bos@lumc.nl (TAB);
m.r.m.jongbloed@lumc.nl
(MRMJ)

Competing interest: The authors declare that no competing interests exist.

**Abstract** Human autonomic neuronal cell models are emerging as tools for modeling diseases such as cardiac arrhythmias. In this systematic review, we compared 33 articles applying 14 different protocols to generate sympathetic neurons and 3 different procedures to produce parasympathetic neurons. All methods involved the differentiation of human pluripotent stem cells, and none employed permanent or reversible cell immortalization. Almost all protocols were reproduced in multiple pluripotent stem cell lines, and over half showed evidence of neural firing capacity. Common limitations in the field are a lack of three-dimensional models and models that include multiple cell types. Sympathetic neuron differentiation protocols largely mirrored embryonic development, with the notable absence of migration, axon extension, and target-specificity cues. Parasympathetic neuron differentiation protocols may be improved by including several embryonic cues promoting cell survival, cell maturation, or ion channel expression. Moreover, additional markers to define parasympathetic neurons in vitro may support the validity of these protocols. Nonetheless, four sympathetic neuron differentiation protocols and one parasympathetic neuron differentiation protocol reported more than two-thirds of cells expressing autonomic neuron markers. Altogether, these protocols promise to open new research avenues of human autonomic neuron development and disease modeling.

## Editor's evaluation

This valuable systematic review provides substantial insights into pluripotent stem cell-derived autonomic postganglionic neuron differentiation techniques. The methodology to collect the underlying evidence is solid. This work provides a helpful resource for researchers who want to establish differentiation pluripotent stem cell into autonomic postganglionic neurons for disease modeling.

## Introduction

The sympathetic and parasympathetic nervous systems, collectively the autonomic nervous system (ANS), typically have peripheral pathways consisting of two serially connected neurons (*Wehrwein et al., 2016*). *Preganglionic* neurons, with somata located in the central nervous system (CNS), form synapses with *postganglionic* neurons with somata located in autonomic ganglia. Sympathetic and parasympathetic postganglionic neurons (henceforth designated 'sympathetic neurons' and 'parasympathetic neurons') communicate directly with effector cells throughout the body to coordinate physiological homeostasis (*Wehrwein et al., 2016*). The widespread distribution of both subdivisions of the ANS in the body is reflected by the myriad of diseases exacerbated or complicated by autonomic dysfunction, including life-threatening cardiac arrhythmias, ventricular remodeling in heart failure, diabetes mellitus, and Parkinson's disease (*Shen and Zipes, 2014*; *Vinik et al., 2003*; *Chen et al., 2020*; *Florea and Cohn, 2014*).

Past ANS research has largely relied on animal models, which are often limited by functional differences with humans (*Zwanenburg et al., 2025*). For instance, under standard laboratory conditions, mice exhibit a predominant sympathetic tone, in contrast to the predominant parasympathetic tone usually observed in humans (*Swoap et al., 2008*). This, combined with ethical concerns and the poor translatability of findings in animal studies to human contexts (*Robinson et al., 2019*), shows the need for valid human ANS models. However, primary human autonomic neurons are difficult to obtain and maintain in culture.

As an alternative, a number of promising sympathetic nervous system models derived from human pluripotent stem cells (hPSCs) have been published (last reviewed in 2019) (*Wu and Zeltner, 2019*). Patient-derived hPSCs have the advantage of offering patient-specific research opportunities. Another approach, conditional immortalization, has previously been used to reversibly induce the proliferation of primary cells with otherwise limited mitotic capacities, such as human epicardium-derived cells and atrial myocytes (*Harlaar et al., 2022*; *Ge et al., 2021*). Cells generated in this manner can regain the functional characteristics of the source cells upon exiting the state of proliferation. In particular, cells derived from the conditional immortalization of autonomic neurons could be especially well-suited for high-throughput applications like drug screening.

However, a number of challenges remain to be addressed in this relatively young field of research. Generally, hPSC-derived cells do not fully recapitulate adult phenotypes, which may limit validity for modeling of adult-onset diseases (*Mateos-Aparicio et al., 2020*; *Karbassi et al., 2020*). Validity for modeling in vivo autonomic neurons may also be increased by incorporating important embryonic signaling cues in autonomic neuron differentiation methods. Another challenge is the considerable proportions of cells other than autonomic neurons generated by many current methods, which may obscure or skew measurements. Additionally, care must be taken to confirm autonomic neuron identities in vitro, outside of the anatomical and physiological context. This is particularly the case for autonomic neurons originating from hPSCs, which can theoretically differentiate towards every cell type of the body, including central cholinergic or noradrenergic neurons (*Thomson et al., 1998*; *Takahashi et al., 2007*).

The aim of this systematic review is to summarize and discuss the currently available methods to derive autonomic neurons. Therefore, the following questions were asked:

1. How do in vitro autonomic neuron differentiation strategies compare to in vivo signaling cues during the embryonic development of the ANS?
2. Which molecular definitions of autonomic neurons are applied in vitro?
3. How efficient are the current differentiation strategies to generate autonomic neurons?
4. Which functional characteristics are shown by autonomic neurons generated in vitro?
5. Which current challenges should future studies address?

## Results

### Study selection

In total, 1440 records were retrieved. After excluding duplicates, and screening titles, abstracts, and full texts, 33 articles were retained (*Figure 1*). Each excluded article during full-text screening is listed in *Supplementary file 1*. The included articles described a total of 14 different protocols to generate sympathetic neurons ('sympathetic protocols', in brief) and 3 different procedures to

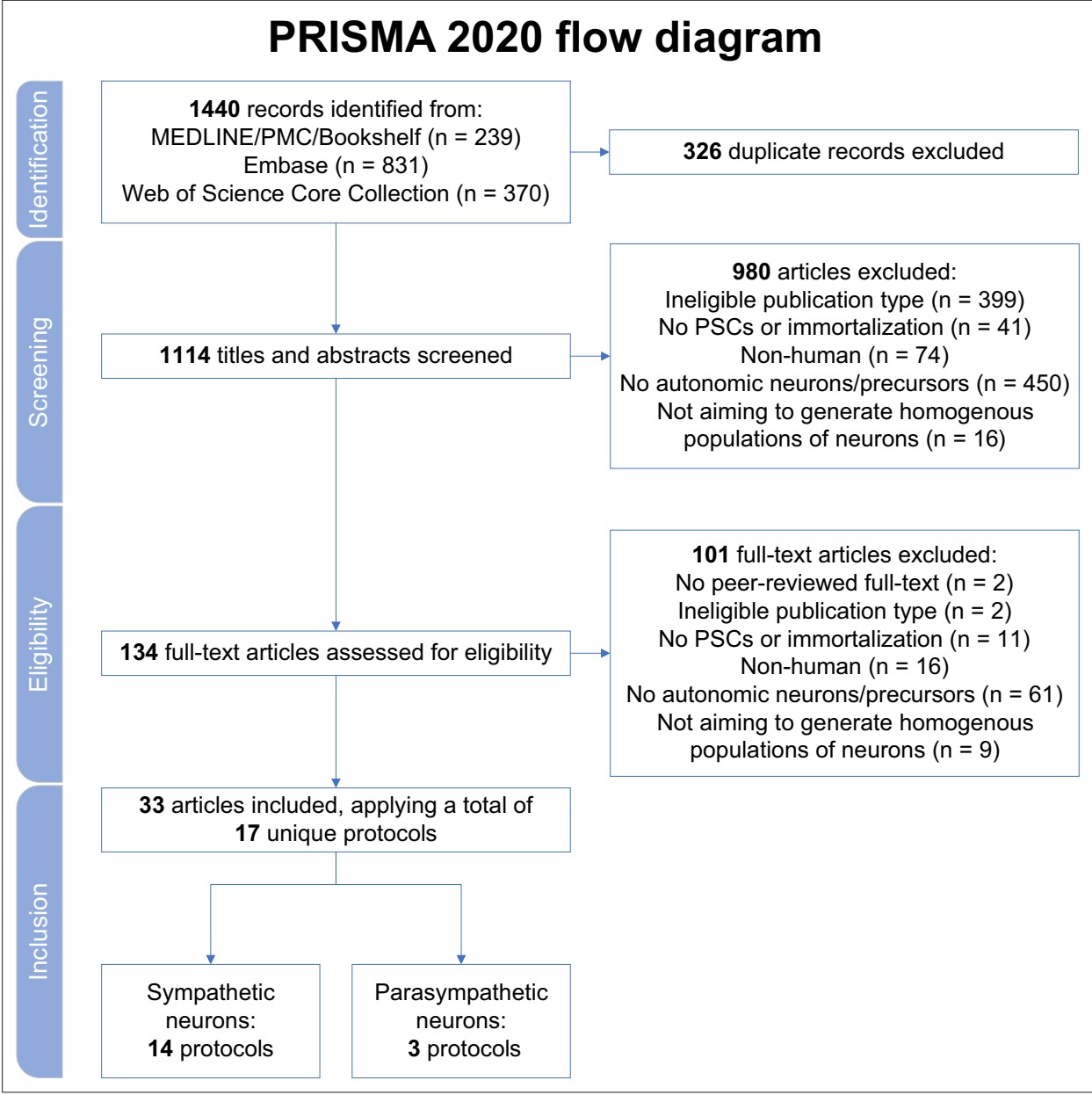

**Figure 1.** Study selection. MEDLINE, Medical Literature Analysis and Retrieval System Online; PMC, PubMed Central; PRISMA, Preferred Reporting Items for Systematic reviews and Meta Analyses; PSCs, pluripotent stem cells.

produce parasympathetic neurons ('parasympathetic protocols', in brief) (*Table 1*; *Huang et al., 2016*; *Cheng et al., 2024*; *Zhang et al., 2016*; *Oh et al., 2016*; *Zeltner et al., 2016*; *Saito-Diaz et al., 2019*; *Frith et al., 2018*; *Frith and Tsakiridis, 2019*; *Saldana-Guerrero et al., 2024*; *Kirino et al., 2018*; *Amer-Sarsour et al., 2024*; *Saleh et al., 2024*; *Carr-Wilkinson et al., 2018*; *Hackland et al., 2019*; *Gomez et al., 2019*; *Wu and Zeltner, 2020*; *Wu et al., 2022a*; *Wu et al., 2022b*; *Wu et al., 2023*; *Wu et al., 2024a*; *Winbo et al., 2020*; *Winbo et al., 2021*; *Bernardin et al., 2022*; *Li et al., 2023*; *Van Haver et al., 2024*; *Fan et al., 2024a*; *Fan et al., 2024b*; *Takayama et al., 2020*; *Takayama et al., 2023*; *Akagi et al., 2024a*; *Akagi et al., 2024b*; *Goldsteen et al., 2022*; *Wu et al., 2024b*). All articles derived neurons from hPSCs, and none applied immortalization techniques. Most articles described protocol development or methods (20/33). The remainder focused exclusively on protocol applications.

**Table 1.** Characteristics of the included articles.

Articles are grouped per protocol and neuron type, in chronological order. Rows in bold indicate the earliest article per protocol.

| Reference | Journal (ISO 4) | Neuron type | Article type | Source cells |
|---|---|---|---|---|
| *Huang et al., 2016* | *Sci Rep* | Sympathetic | Protocol development | PSCs |
| *Cheng et al., 2024*[*] | *Sci Rep* | Sympathetic | Protocol application | iPSCs |
| *Zhang et al., 2016* | *PLoS One* | Sympathetic | Protocol development/application | ESCs |
| *Oh et al., 2016* | *Cell Stem Cell* | Sympathetic | Protocol development | PSCs |
| *Zeltner et al., 2016* | *Nat Med* | Sympathetic | Protocol development/application | PSCs |
| *Saito-Diaz et al., 2019* | *Curr Protoc Stem Cell Biol* | Sympathetic | Methodological | PSCs |
| *Frith et al., 2018* | *eLife* | Sympathetic | Protocol development | PSCs |
| *Frith and Tsakiridis, 2019* | *Curr Protoc Stem Cell Biol* | Sympathetic | Methodological | PSCs |
| *Saldana-Guerrero et al., 2024* | *Nat Commun* | Sympathetic | Protocol application | ESCs |
| *Kirino et al., 2018* | *Sci Rep* | Sympathetic | Protocol development | PSCs |
| *Amer-Sarsour et al., 2024* | *Embo J* | Sympathetic | Protocol application | iPSCs |
| *Saleh et al., 2024* | *Free Radic Biol Med* | Sympathetic | Protocol application | iPSCs |
| *Carr-Wilkinson et al., 2018* | *Stem Cells Int* | Sympathetic | Protocol development | ESCs |
| *Hackland et al., 2019* | *Stem Cell Reports* | Sympathetic | Protocol development/application | PSCs |
| *Gomez et al., 2019* | *Development* | Sympathetic | Protocol development/application | PSCs |
| *Wu and Zeltner, 2020* | *J Vis Exp* | Sympathetic | Methodological | ESCs |
| *Wu et al., 2022a* | *Clin Auton Res* | Sympathetic | Protocol application | PSCs |
| *Wu et al., 2022b* | *Nat Commun* | Sympathetic | Protocol development/application | PSCs |
| *Wu et al., 2023* | *Front Neurosci* | Sympathetic | Protocol application | ESCs |
| *Wu et al., 2024a* | *STAR Protoc* | Sympathetic | Methodological | ESCs |
| *Winbo et al., 2020* | *Am J Physiol Heart Circ Physiol* | Sympathetic | Protocol development | iPSCs |
| *Winbo et al., 2021* | *Am J Physiol Heart Circ Physiol* | Sympathetic | Protocol application | iPSCs |
| *Bernardin et al., 2022* | *Cells* | Sympathetic | Protocol application | iPSCs |
| *Li et al., 2023* | *Philos Trans R Soc Lond B Biol Sci* | Sympathetic | Protocol application | iPSCs |
| *Van Haver et al., 2024* | *iScience* | Sympathetic | Protocol development/application | PSCs |
| *Fan et al., 2024a* | *J Mol Neurosci* | Sympathetic | Protocol development | PSCs |
| *Fan et al., 2024b* | *Cell Rep Med* | Sympathetic | Protocol application | PSCs |
| *Takayama et al., 2020* | *Sci Rep* | Sympathetic or parasympathetic | Protocol development | PSCs |
| *Takayama et al., 2023* | *Int J Mol Sci* | Sympathetic or parasympathetic | Protocol application | PSCs |
| *Akagi et al., 2024a* | *FEBS Open Bio* | Parasympathetic | Protocol application | iPSCs |
| *Akagi et al., 2024b* | *Molecules* | Sympathetic or parasympathetic | Protocol application | iPSCs |
| *Goldsteen et al., 2022* | *Front Pharmacol* | Parasympathetic | Protocol development | ESCs |
| *Wu et al., 2024b* | *Cell Stem Cell* | Parasympathetic | Protocol development/application | PSCs |

ESCs, embryonic stem cells; iPSCs, induced pluripotent stem cells; ISO, International Organization for Standardization; PSCs, pluripotent stem cells (iPSCs or ESCs).

*****Cheng et al., 2024** applied three of the protocols included in this review, by **Huang et al., 2016**, **Frith et al., 2018**, and **Kirino et al., 2018**.

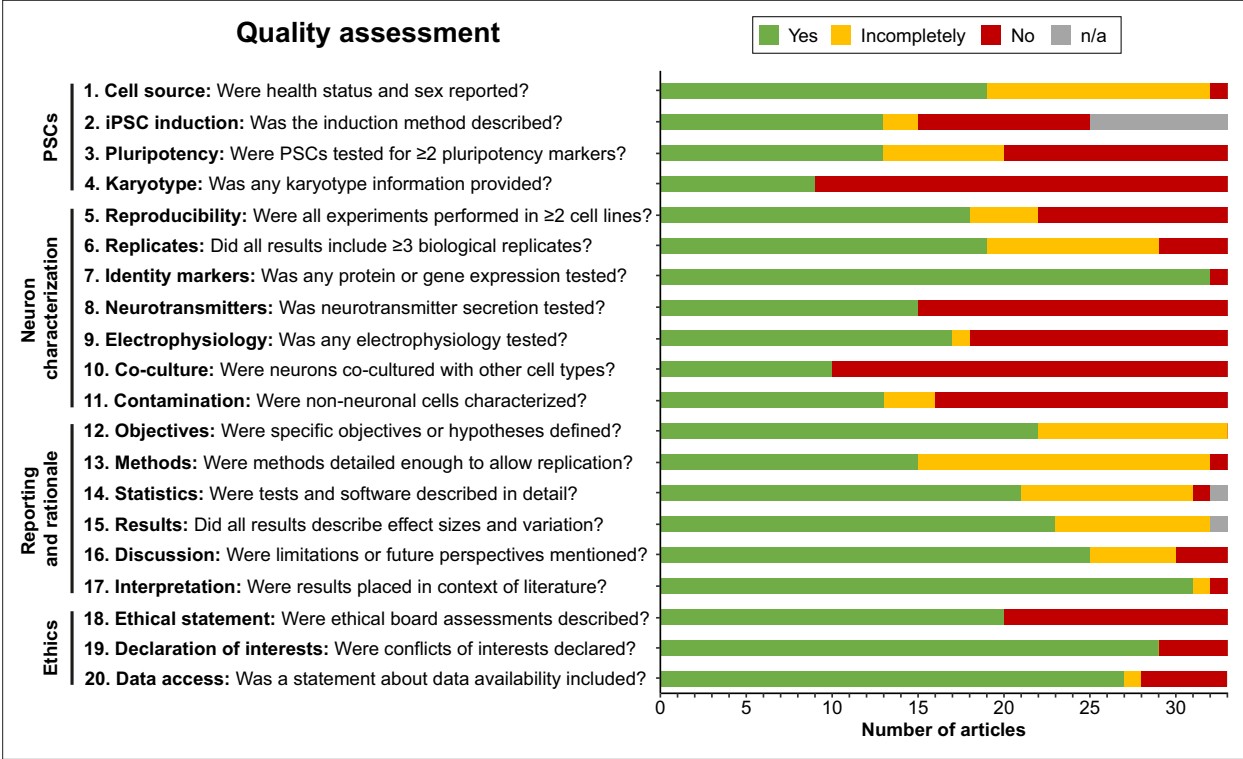

**Figure 2.** Quality assessment. Quality assessment results per criterion. Criteria topics are indicated to the left of the criteria. See *Figure 2—figure supplement 1* for results per article, and *Supplementary file 2* for detailed criteria. iPSCs, Induced pluripotent stem cells; n/a, not applicable; PSCs, pluripotent stem cells.

The online version of this article includes the following figure supplement(s) for figure 2:

**Figure supplement 1.** Quality assessment results per article.

## Quality assessment

Next, articles were judged by two independent reviewers based on the criteria formulated in *Supplementary file 2*. Interrater reliability was high (Cohen's kappa=0.914). Generally, articles performed well on reporting and rationale items, whereas information about PSC-related items, such as induction methods, pluripotency markers, and karyotype validation, was often incomplete or absent (*Figure 2*). When pooling articles per protocol, all protocols were reproduced in multiple PSC lines, except for the protocols by *Zhang et al., 2016*; *Carr-Wilkinson et al., 2018*; and *Goldsteen et al., 2022*. Furthermore, nearly all articles investigated neuronal identity markers via gene or protein expression (32/33). However, functional outcomes such as neurotransmitter secretion (15/33), electrophysiology (18/33), and co-culture with other cell types (10/33) were reported markedly less frequently. Almost a third of articles (10/33) reported no functional outcomes (*Figure 2—figure supplement 1*). Additionally, only half of the articles reported identities of contaminating cell types (i.e., cells other than autonomic neurons) (16/33).

## How do in vitro autonomic neuron differentiation strategies compare to in vivo signaling cues during the embryonic development of the autonomic nervous system?

During embryonic development, autonomic neurons arise from neural crest cells (NCCs), a multipotent migratory cell population, which, in turn, is derived from the neural plate border. The neural plate border emerges between the neural plate and non-neural ectoderm, which respectively form the CNS and epidermis (*Thawani and Groves, 2020*). Distinct neural crest and preplacodal regions then form at the neural plate border (*Thawani and Groves, 2020*). Next, NCCs migrate throughout the embryo in a rostrocaudal sequence as the neural plate folds to form the neural tube. Depending on their point of origin, NCCs generate different cell types (*Martik and Bronner, 2017*). Postganglionic sympathetic

neurons mainly originate from trunk NCCs (*Martik and Bronner, 2017*), whereas postganglionic parasympathetic neurons arise mostly from more rostrally situated vagal NCCs (*Espinosa-Medina et al., 2014*).

A portion of the trunk NCCs migrate ventrally to arrive at the dorsal aorta, where they coalesce to form the sympathetic ganglia (*Rickmann et al., 1985*; *Kasemeier-Kulesa et al., 2006*). Sympathetic neurons project to target organs from this paravertebral position, in contrast to parasympathetic somata, which are situated in close proximity to their target organs. Reflecting this, vagal NCCs fated to generate parasympathetic ganglia migrate along the course of preganglionic cranial nerves towards their target organs as Schwann cell precursors (SCPs) (*Espinosa-Medina et al., 2014*; *Verberne et al., 1998*; *Hildreth et al., 2008*; *Dyachuk et al., 2014*). As autonomic ganglia mature, mitotic precursor cells differentiate to postmitotic neurons, which rely on target-derived factors for survival (*Gonsalvez et al., 2013*; *Koszinowski et al., 2015*).

In the following section, the different in vitro strategies have been compared to each stage of the autonomic neuronal development outlined above. Most knowledge on signaling requirements during embryonic development is based on a variety of non-human vertebrate models. To most closely approach the human situation, we based our representation of in vivo signaling on studies performed in amniote models, namely mammals and birds.

## Neural crest cell induction

Most protocols initiated differentiation by inducing NCCs. Based on common signaling cues during NCC induction, we grouped protocols into four categories (*Figure 3*). One of the earliest approaches, applied in two sympathetic protocols (*Zhang et al., 2016*; *Carr-Wilkinson et al., 2018*), utilized the stromal cell-derived inducing activity (SDIA) of PA6 cells, a mouse preadipocyte cell line (*Kawasaki et al., 2000*). Although both protocols reported the expression of catecholaminergic markers, sympathetic neuron yields were low, despite the use of fluorescence-activated cell sorting (FACS) to select for NCC markers during differentiation. Moreover, the signaling cues involved in SDIA have not been fully defined and can also induce dopaminergic neuron differentiation (*Schwartz et al., 2012*). Together, this suggests SDIA does not specifically recapitulate the embryonic development of sympathetic neurons.

All other approaches applied activation of Wnt signaling via CHIR99021 (CHIR)-based glycogen synthesis kinase 3 inhibition to generate NCCs (*Figure 3*, for additional culture details, see *Supplementary file 3*). The modes of action for all small molecules used in the included protocols are provided in *Supplementary file 4*. Activation of Wnt signaling via CHIR in these protocols is in line with avian NCC development, which also depends on Wnt signaling (*Figure 4A*; *Patthey et al., 2009*). Of protocols applying CHIR, two parasympathetic protocols (*Takayama et al., 2020*; *Goldsteen et al., 2022*) and four sympathetic protocols (*Oh et al., 2016*; *Zeltner et al., 2016*; *Winbo et al., 2020*; *Takayama et al., 2020*) applied a technique coined 'dual SMAD inhibition'. These protocols combined SMAD2/3 inhibition and SMAD1/5/8 inhibition from the start of differentiation, and added CHIR from day 2 of differentiation. Transforming growth factor beta (TGFβ)-, Activin-, and Nodal-specific SMAD2/3 signaling was always inhibited via anaplastic lymphoma kinase (ALK)4/5/7 receptor inhibitor SB431542 (SB) (*Inman et al., 2002*), and bone morphogenetic protein (BMP)-specific SMAD1/5/8 signaling was inhibited by small molecule inhibitors LDN193189 (LDN) or dorsomorphin (DMH) (*Chambers et al., 2009*; *Chambers et al., 2012*). Although little seems to be known about SMAD2/3 signaling requirements for NCC induction in vivo, SMAD1/5/8 signaling is inhibited at the start of neural crest specification, followed by activation as neural crest development progresses (*Patthey et al., 2009*; *Faure et al., 2002*). This may parallel the use of LDN or DMH. Furthermore, in the absence of CHIR, dual SMAD inhibition instead results in high proportions of neural plate marker expression in vitro (*Chambers et al., 2009*). This resembles the way inhibition of Wnt induces neural plate expression in avian ectodermal explants, which would otherwise express neural crest markers (*Figure 4A*; *Patthey et al., 2009*).

The addition of CHIR to dual SMAD inhibition results in ~70% of cells expressing neural crest marker SRY-box transcription factor 10 (SOX10) by day 12 of differentiation (*Chambers et al., 2012*). Addition of two other small molecules (DAPT, an indirect Notch signaling inhibitor; *Geling et al., 2002*), and SU5402, an inhibitor of fibroblast growth factor receptor 1 (FGFR1) and vascular endothelial growth factor receptor 2 (VEGFR2) (*Mohammadi et al., 1997*), slightly increases the proportion

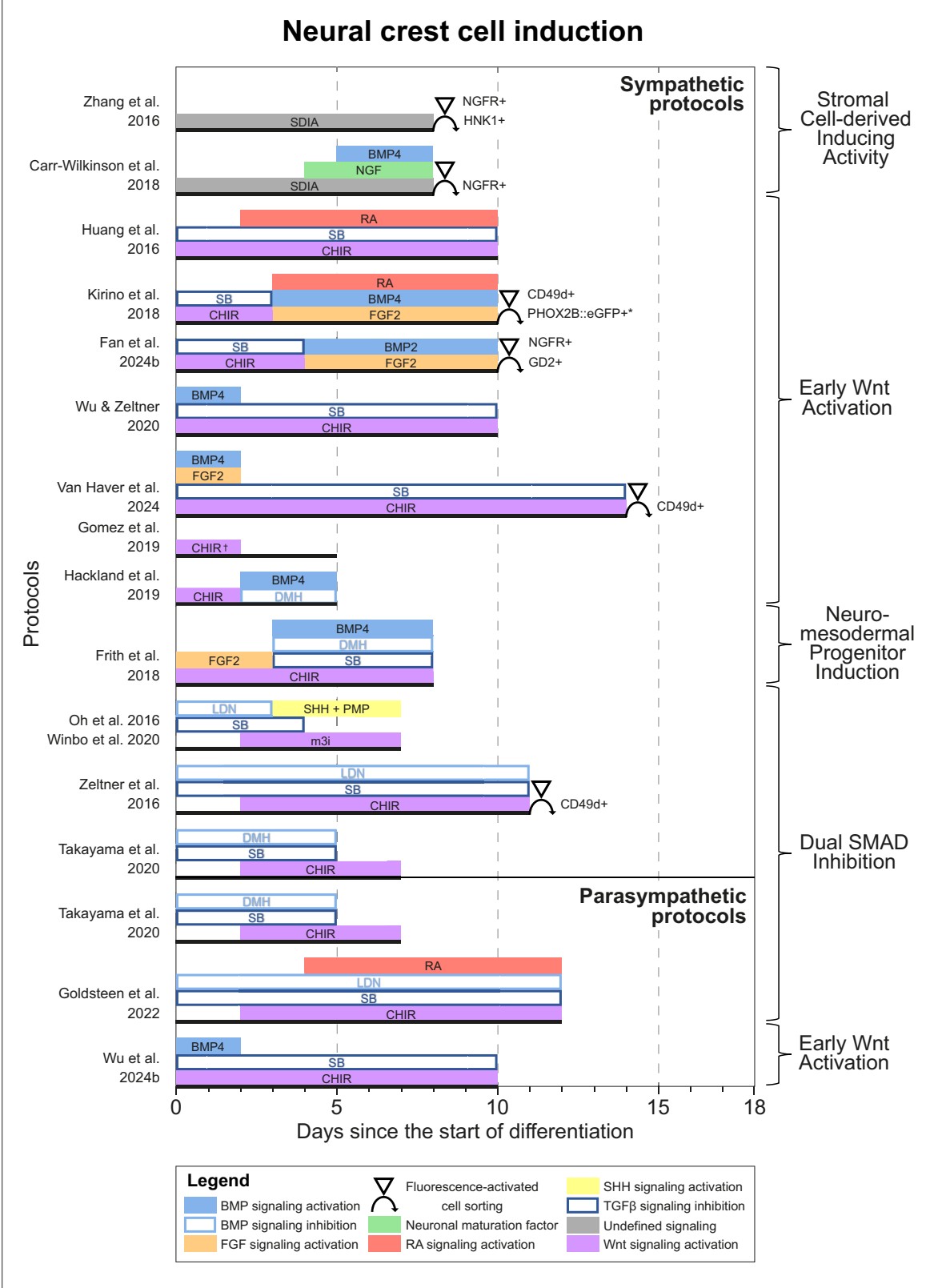

**Figure 3.** Neural crest cell induction. Timings and signaling cues used during the first phase of differentiation until neural crest induction per unique protocol. Duration of this phase per protocol is indicated by the horizontal black bars. Categories of similar approaches are indicated to the right of the figure. Molecules targeting similar pathways have been grouped by color. Colors also match the signaling cues in **Figures 4–6**. * Selection step yields optimal cell purity, but this is not required. † **Gomez et al., 2019** identified two optimal CHIR concentrations for neural crest induction, 3 µM and

*Figure 3 continued on next page*

*Figure 3 continued*

10 µM. 10 µM was used for sympathetic neuron differentiation. BMP, bone morphogenetic protein; CD49d, Integrin subunit α4; CHIR, CHIR99021; DMH, dorsomorphin; eGFP, enhanced green fluorescent protein; FGF, fibroblast growth factor; GD2, disialoganglioside; HNK1, human natural killer-1; LDN, LDN193189; m3i, Modified three inhibitor approach (CHIR99021, DAPT, and PD173074); NGF, nerve growth factor; NGFR, nerve growth factor receptor; PHOX2B, paired-like homeobox 2b; PMP, purmorphamine; RA, retinoic acid; SB, SB431542; SDIA, stromal cell-derived inducing activity; SHH, Sonic hedgehog; TGFβ, transforming growth factor beta.

of cells expressing SOX10 by day 12 of differentiation to ~80% (*Chambers et al., 2012*). Oh et al. later modified this approach (i.e., modified three inhibitor approach) by substituting SU5402 for PD173074 (*Oh et al., 2016*), another FGFR inhibitor. The mechanisms by which SU5402 or PD173074, and DAPT contribute to NCC induction are still unknown. Although Notch signaling influences fate decisions after NCC induction, any direct role of Notch during NCC induction is unclear in chicken and mice (*Stuhlmiller and García-Castro, 2012a*). Furthermore, fibroblast growth factor (FGF) signaling inhibition by SU5402 or PD173074 seems to contradict the FGF requirement for NCC formation in vivo (*Stuhlmiller and García-Castro, 2012b*; *Figure 4A*). On the other hand, the concentration of PD173074 used, 0.2 µM, may not entirely suppress endogenous cellular FGF signaling (*Gomez et al., 2019*).

CHIR-mediated Wnt signaling activation from day 0 of differentiation instead of day 2 of differentiation dramatically reduces SOX10-positive NCC induction under dual SMAD inhibition conditions (*Mica et al., 2013*). Therefore, we consider protocols employing CHIR from the start of differentiation (i.e., 'early Wnt activation') to be distinct from dual SMAD inhibition protocols. Protocols in the 'early Wnt activation' category often inhibited SMAD1/5/8 signaling via SB (*Huang et al., 2016*; *Kirino et al., 2018*; *Hackland et al., 2019*; *Wu and Zeltner, 2020*; *Van Haver et al., 2024*; *Fan et al., 2024a*; *Wu et al., 2024b*), but did not inhibit BMP-specific SMAD2/3 signaling. The lack of BMP modulation in most early Wnt activation protocols seemingly contradicts evidence from embryonic development (*Patthey et al., 2009*; *Faure et al., 2002*). Moreover, similar to dual SMAD inhibition protocols, almost no early Wnt activation protocols actively stimulate FGF signaling simultaneously with Wnt signaling stimulation, both of which are required for avian NCC development (*Stuhlmiller and García-Castro, 2012b*). Nonetheless, three of the four protocols retrieved by our query, which reported high sympathetic neuron differentiation purities (≥67% of cells expressing tyrosine hydroxylase (TH)), relied on a form of early Wnt activation without active FGF stimulation for NCC induction (*Kirino et al., 2018*; *Wu and Zeltner, 2020*; *Fan et al., 2024a*). Information on endogenous cellular Wnt, BMP, and FGF signaling in these protocols would help clarify this apparent discrepancy.

A final induction strategy applied by *Frith et al., 2018* is based on neuromesodermal progenitor (NMP) induction. Although the evidence is not conclusive, NMPs may contribute to posterior NCCs in vivo (*Figure 4B*), as suggested by various grafting and lineage-tracing studies (*Shaker et al., 2021*; *Rodrigo Albors et al., 2018*; *Zhao et al., 2007*; *Javali et al., 2017*). Most relevant to sympathetic neurons, NMPs are located in the tail bud during axial elongation (*Olivera-Martinez et al., 2012*; *Wymeersch et al., 2016*). As the name implies, NMPs express the pro-mesodermal marker, T-box transcription factor T (TBXT), and the neuroepithelial marker, SRY-box transcription factor 2 (SOX2). In the protocol developed by *Frith et al., 2018*, >80% of cells co-expressed these markers after 3 days of concomitant Wnt and FGF2 signaling activation. Indeed, both these signaling pathways are also upregulated in vivo in the primitive streak where NMPs can be found (*Wymeersch et al., 2019*; *Figure 4B*). The involvement of FGF and Wnt signaling in NMP induction is also supported by transcriptome analyses (*Amin et al., 2016*; *Gouti et al., 2017*). After generating NMPs, Frith et al. induced NCCs via 'top-down BMP inhibition' (*Hackland et al., 2017*), resulting in around 60% SOX10-positive cells.

## Caudalization

NMP induction, as described above, reflects broader efforts to achieve posterior or trunk NCC identity (*Figure 4—figure supplement 1*). However, early protocols for NCC induction mostly generated NCCs of anterior identity (*Huang et al., 2016*). On the other hand, the majority of SOX10-positive NCCs generated via NMP induction expressed homeobox C9 (HOXC9) (*Frith et al., 2018*), which indicates the trunk position (*Dasen et al., 2003*). HOXC9 expression can also depend on CHIR concentration, as shown by *Gomez et al., 2019*. By comparing 2-day CHIR exposures of different

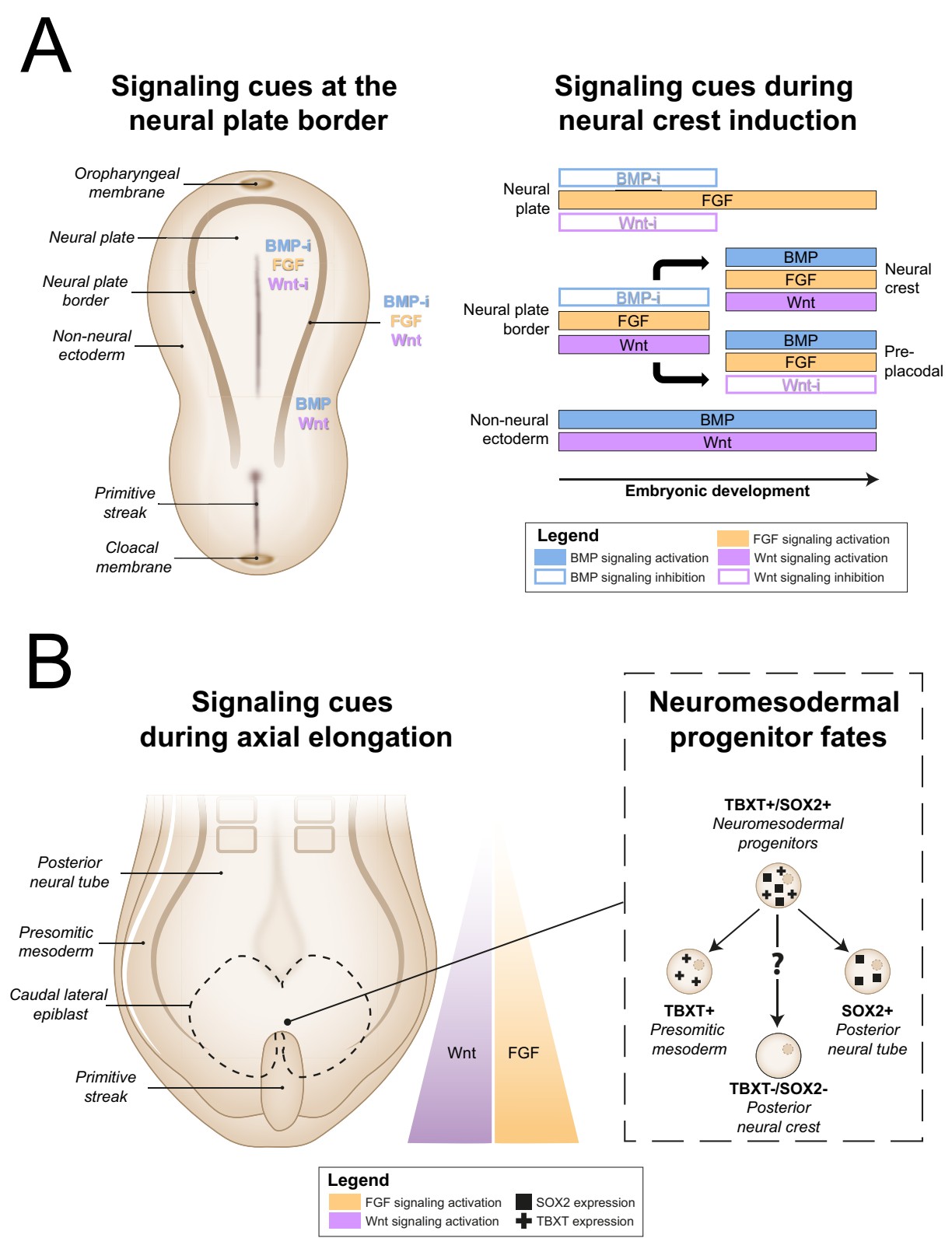

**Figure 4.** In vivo neural crest induction signaling requirements. (**A**) General signaling requirements for distinct populations at the neural plate border. Left, a dorsal schematic view of the embryo and signaling cues (indicated in colored text) present near the neural plate border during early gastrulation are shown. Right, the temporal sequence of signaling cues required for distinct populations near the neural plate border in amniotes between gastrulation and neurulation is shown (based on *Thawani and Groves, 2020*). (**B**) Neuromesodermal progenitors arise in the tailbud

*Figure 4 continued on next page*

*Figure 4 continued*

during axial elongation under conditions of high Wnt and FGF signaling activation. Wnt and FGF concentrations form a rostrocaudal gradient, with highest concentrations in the tailbud. Right, neuromesodermal progenitors possibly contribute to posterior neural crest cell populations. BMP, bone morphogenetic protein; BMP-i, BMP signaling inhibition; FGF, fibroblast growth factor; SOX2, SRY-box transcription factor 2; TBXT, T-box transcription factor T; Wnt-i, Wnt signaling inhibition.

The online version of this article includes the following figure supplement(s) for figure 4:

**Figure supplement 1.** Rostrocaudal boundaries of vagal and trunk neural crest.

---

concentrations between 0 and 12 µM, CHIR concentrations of 3 µM and 10 µM were found to most efficiently induce SOX10 and paired box 7 (PAX7) expression. Strikingly, cells treated with 10 µM CHIR showed vastly higher *HOXC9* expression than cells treated with 3 µM CHIR. Further investigation is warranted to explore whether this relatively simple approach to generate posterior NCCs can be used to efficiently generate mature sympathetic neurons.

Another tactic to induce posterior gene expression, often referred to as caudalization, utilizes retinoic acid (RA), sometimes combined with activation of Wnt signaling (*Wichterle et al., 2002*; *Irioka et al., 2005*; *Lippmann et al., 2015*). Three autonomic protocols applied this technique to stimulate *HOX* gene expression as far posterior as *HOXC9* and homeobox B9 (*HOXB9*) at the population level (*Huang et al., 2016*; *Kirino et al., 2018*; *Goldsteen et al., 2022*). However, there is controversy about the efficiency by which RA can induce posterior markers. In one experiment, RA-based caudalization induced posterior *HOX* gene expression much less efficiently than NMP induction (*Frith et al., 2018*). Additionally, in the work by Gomez et al., RA mainly induced the anterior marker homeobox B4 (*HOXB4*), but not *HOXC9* (*Gomez et al., 2019*). Altogether, considering the instructive role *HOX* genes have during development (*Mallo et al., 2010*), we recommend demonstrating posterior *HOX* gene expression when establishing new sympathetic protocols.

## Sympathetic neurogenesis

After NCC induction, many sympathetic protocols (11/14) (*Huang et al., 2016*; *Oh et al., 2016*; *Frith et al., 2018*; *Kirino et al., 2018*; *Carr-Wilkinson et al., 2018*; *Hackland et al., 2019*; *Gomez et al., 2019*; *Winbo et al., 2020*; *Van Haver et al., 2024*; *Fan et al., 2024a*; *Takayama et al., 2020*) employed signaling cues encountered by migratory NCCs en route to the dorsal aorta, such as BMPs (10/14) (*Huang et al., 2016*; *Oh et al., 2016*; *Frith et al., 2018*; *Kirino et al., 2018*; *Carr-Wilkinson et al., 2018*; *Hackland et al., 2019*; *Winbo et al., 2020*; *Van Haver et al., 2024*; *Fan et al., 2024a*; *Takayama et al., 2020*), and sonic hedgehog (SHH) (6/14) (*Oh et al., 2016*; *Frith et al., 2018*; *Hackland et al., 2019*; *Gomez et al., 2019*; *Winbo et al., 2020*; *Van Haver et al., 2024*; *Figure 5A*, full-length sympathetic protocol overviews are shown in *Figure 5—figure supplement 1*). This mostly involved the use of BMP4, and a combination of recombinant SHH and the SHH agonist purmorphamine (PMP). In vivo, NCCs migrate ventrally towards the dorsal aorta (*O'Rahilly and Müller, 2007*; *Betters et al., 2010*; *Bronner-Fraser, 1986*; *Serbedzija et al., 1990*; *Figure 5B*), attracted by neuregulin 1 (NRG1) and C-X-C motif chemokine ligand 12 (CXCL12) (*Saito et al., 2012*), and repelled by semaphorin 3A (SEMA3A) (*Kawasaki et al., 2002*). Upon arrival, NCCs differentiate into sympathetic precursors under the influence of BMP4 and BMP7 emitted by the dorsal aorta (*Saito et al., 2012*; *Reissmann et al., 1996*; *Shah et al., 1996*; *Schneider et al., 1999*; *Figure 5B*). In vitro, the only sympathetic protocol to investigate the sympathetic neuron differentiation efficiency of different BMPs did not conclusively show any of BMP2, BMP4, and BMP7 to be superior to the others (*Huang et al., 2016*). In any case, three of the four sympathetic protocols reporting high differentiation efficiencies (≥67% of cells expressing TH) actively stimulated BMP signaling (*Kirino et al., 2018*; *Winbo et al., 2020*; *Fan et al., 2024a*), suggesting this is also an important component of sympathetic neuron differentiation in vitro.

SHH may also participate in sympathetic differentiation, as suggested by its emission along the NCC migratory path at the notochord and floorplate of the neural tube (*Echelard et al., 1993*; *Danesin and Soula, 2017*). In vitro, exposure to SHH increases the proportion of cells expressing TH in populations of primary NCCs or sympathetic neurons (*Reissmann et al., 1996*; *Williams et al., 2000*). Further supporting this notion, *Shh*-null mice show dysmorphic hypoplastic sympathetic ganglia with delayed neuronal development (*Morikawa et al., 2009*). Additionally, proximity of the notochord and floorplate to the dorsal aorta is required for normal sympathetic neuronal development (*Stern et al.,*

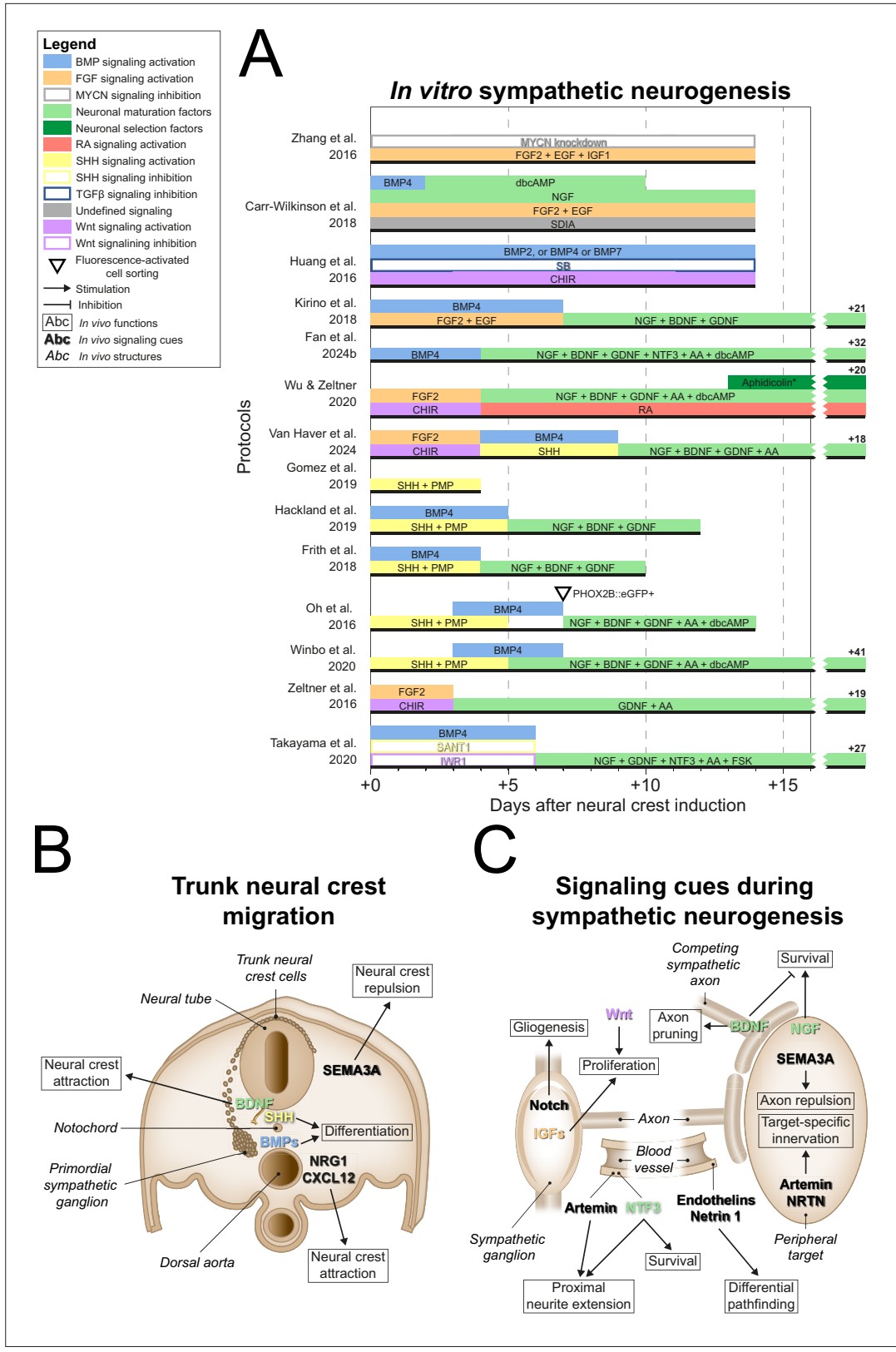

**Figure 5.** Sympathetic neurogenesis. (**A**) Timings and signaling cues used from neural crest induction until the end of sympathetic neuron differentiation. Duration of this phase per protocol is indicated by the horizontal black bars. Total durations of this phase exceeding the width of the graph are indicated to the right of the graph. Molecules targeting similar pathways have been grouped by color. (**B**) Transverse cross section of the

*Figure 5 continued on next page*

*Figure 5 continued*

trunk of an embryo during neural crest migration. Signaling requirements for ventral neural crest migration and sympathetic specification are indicated by bold text. Signaling cues targeted by the protocols in (**A**) are indicated with colored text matching those in the figure legend. (**C**) Schematic view of the signaling requirements for sympathetic precursor proliferation and target innervation. The discontinuous axon and blood vessel represent the large distance from the sympathetic ganglia to their peripheral targets. * Aphidicolin selection yields optimal cell purity. However, this is not required. AA, ascorbic acid; BDNF, brain-derived neurotrophic factor; BMP, bone morphogenetic protein; CHIR, CHIR99021; CXCL12, C-X-C motif chemokine ligand 12; dbcAMP, dibutyryl cyclic adenosine monophosphate; EGF, epidermal growth factor; eGFP, enhanced green fluorescent protein; FGF, fibroblast growth factor; FSK, forskolin; GDNF, glial cell line derived neurotrophic factor; IGF, insulin-like growth factor; MYCN, MYCN proto-oncogene; NGF, nerve growth factor; NRG1, neuregulin 1; NRTN, neurturin; NTF3, neurotrophin 3; PHOX2B, paired-like homeobox 2b; PMP, purmorphamine; RA, retinoic acid; SEMA3A, semaphorin 3A; SB, SB431542; SDIA, stromal cell-derived-inducing activity; SHH, Sonic hedgehog; TGFβ, transforming growth factor beta.

The online version of this article includes the following figure supplement(s) for figure 5:

**Figure supplement 1.** Total overview of all sympathetic protocols.

---

*1991*). However, direct stimulation of SHH signaling is only included in one of the four sympathetic protocols reporting high efficiencies (≥67% TH-positive) (*Winbo et al., 2020*). At most, this suggests that active stimulation of SHH signaling plays an accessory role in vitro.

Upon formation at the dorsal aorta, the primordial sympathetic ganglia consist mainly of proliferative cells (*Gonsalvez et al., 2013*). In vitro, eight autonomic protocols stimulated the proliferation of NCCs by expansion in low-adherence conditions via FGF2 and/or epidermal growth factor (EGF) (*Zhang et al., 2016*; *Zeltner et al., 2016*; *Kirino et al., 2018*; *Wu and Zeltner, 2020*; *Van Haver et al., 2024*; *Fan et al., 2024a*; *Goldsteen et al., 2022*; *Wu et al., 2024b*; *Figure 5A*). Four protocols in this review combined these factors with activation of Wnt signaling (*Zeltner et al., 2016*; *Wu and Zeltner, 2020*; *Van Haver et al., 2024*; *Goldsteen et al., 2022*). This may be analogous to the requirement of canonical Wnt signaling for mitotic sympathetic precursor cell maintenance (*Armstrong et al., 2011*; *Figure 5C*). Additionally, Zhang et al. applied IGF1 in this phase (*Zhang et al., 2016*), which supports sympathetic precursor proliferation and neurite outgrowth (*Zackenfels et al., 1995*). At this stage, sympathetic precursors differentiate into glial and neuronal cells depending on the stimulation or inhibition of Notch signaling, respectively (*Tsarovina et al., 2008*; *Shtukmaster and Huber, 2023*). Yet, no sympathetic protocol included inhibition of Notch signaling after NCC induction. In order to fully recapitulate sympathetic embryonic development, we believe a protocol should be capable of generating both glial cells and neurons from the same progenitor population, depending on Notch signaling. However, generally, the presence of glial cells was not investigated in current protocols.

During the maturation of sympathetic neurons in vivo, neurite development and target innervation depend on various autocrine and target-derived growth factors (*Figure 5C*, and reviewed thoroughly elsewhere; *Scott-Solomon et al., 2021*). Of all sympathetic protocols, four did not proceed past the precursor phase (*Huang et al., 2016*; *Zhang et al., 2016*; *Gomez et al., 2019*; *Van Haver et al., 2024*). Nearly all others employed nerve growth factor (NGF) (9/10) (*Oh et al., 2016*; *Frith et al., 2018*; *Kirino et al., 2018*; *Carr-Wilkinson et al., 2018*; *Hackland et al., 2019*; *Wu and Zeltner, 2020*; *Winbo et al., 2020*; *Takayama et al., 2020*) for neuronal maturation, which stimulates neurite outgrowth and sympathetic neuron survival upon axonal contact with target tissue (*Kuruvilla et al., 2004*; *Crowley et al., 1994*; *Shelton and Reichardt, 1984*). Before this, neurite outgrowth is promoted by the vasculature-derived factors artemin and neurotrophin 3 (NTF3), of which the latter also supports survival (*Kuruvilla et al., 2004*; *Enomoto et al., 1998*; *Honma et al., 2002*; *Verdi and Anderson, 1994*). Artemin, as well as factors like neurturin (NRTN), endothelins, and netrin 1, coordinate differential pathfinding and target-specific sympathetic innervation (*Furlan et al., 2016*; *Manousiouthakis et al., 2014*; *Makita et al., 2008*; *Brunet et al., 2014*). Of all these factors, only NTF3 featured in two sympathetic protocols (*Fan et al., 2024a*; *Takayama et al., 2020*). However, simultaneous NGF application in these protocols likely reduced the effect of NTF3 (*Verdi and Anderson, 1994*).

Instead, most sympathetic protocols which proceeded past the precursor phase (8/10) (*Oh et al., 2016*; *Frith et al., 2018*; *Kirino et al., 2018*; *Hackland et al., 2019*; *Wu and Zeltner, 2020*; *Winbo et al., 2020*; *Van Haver et al., 2024*; *Fan et al., 2024a*) combined NGF with brain-derived

neurotrophic factor (BDNF) and glial cell line-derived neurotrophic factor (GDNF) to promote further neurogenic differentiation (*Figure 5A*). The exact contributions of the latter two factors in these protocols were not reported. In vivo, BDNF secreted by preganglionic axons likely guides primary sympathetic ganglia to their secondary paravertebral position (*Kasemeier-Kulesa et al., 2015*), although differential growth also plays a role (*Kruepunga et al., 2021*). Secondly, BDNF secreted by more mature sympathetic neurons induces axon pruning and cell death in adjacent neurons (*Singh et al., 2008*; *Escudero et al., 2019*; *Figure 5C*). Conversely, GDNF may promote cell proliferation or survival, as suggested by reduced numbers of sympathetic neurons in *Gdnf*-knockout mice (*Moore et al., 1996*). However, the mechanism remains unclear, as knockout of receptor tyrosine kinase *Ret*, required for GDNF family ligand signaling, or co-receptor GDNF family receptor α1, required for preferential GDNF signaling (*Baloh et al., 2000*), do not fully replicate this phenotype (*Enomoto et al., 1998*; *Durbec et al., 1996*).

## Parasympathetic neurogenesis

After generating NCCs, each parasympathetic protocol employed divergent tactics to generate parasympathetic precursors (*Figure 6A*, full-length parasympathetic protocol overviews are shown in *Figure 6—figure supplement 1*). In vivo, vagal NCCs migrate along preganglionic axons to the site of the prospective parasympathetic ganglia in the form of SCPs, which depend on axonal signals like NRG1 for survival (*Solovieva and Bronner, 2021*; *Dong et al., 1995*; *Newbern and Birchmeier, 2010*; *Figure 6B*). Goldsteen et al. and Wu et al. showed increased expression of axial markers of vagal NCCs (*Goldsteen et al., 2022*; *Wu et al., 2024b*), homeobox B3 (*HOXB3*) and homeobox B5 (*HOXB5*). However, only Wu et al. explicitly targeted SCP generation via NRG1 in combination with CHIR and FGF2 (*Wu et al., 2024b*). In strong support of the recapitulation of embryonic development in this protocol, Wu et al. showed that the resulting SOX10-positive SCPs were also capable of Schwann cell generation.

In vivo, the development of ciliary ganglion neurons, a parasympathetic neuron subtype, depends on local BMPs, possibly BMP4, BMP5, and/or BMP7 (*Müller and Rohrer, 2002*). The only parasympathetic protocol to actively stimulate BMP signaling after NCC induction was the protocol by *Takayama et al., 2020*. As mentioned before, sympathetic ganglia also rely on BMPs during development (*Saito et al., 2012*; *Reissmann et al., 1996*; *Shah et al., 1996*; *Schneider et al., 1999*). Instead of SCP generation, this protocol inhibited alternative NCC fates to retain progenitor populations capable of both sympathetic and parasympathetic differentiation. SHH signaling stimulates enteric neuron-fated NCC proliferation (*Fu et al., 2004*), and constitutively active Wnt signaling abolishes autonomic neuron differentiation in favor of sensory neuron differentiation (*Lee et al., 2004*). Therefore, the inhibition of alternative NCC fates was implemented by SHH and Wnt inhibition, via SANT1 and IWR1, respectively (*Takayama et al., 2020*). This strategy contrasts strikingly with the majority of other autonomic protocols (9/12) (*Huang et al., 2016*; *Oh et al., 2016*; *Zeltner et al., 2016*; *Frith et al., 2018*; *Hackland et al., 2019*; *Gomez et al., 2019*; *Wu and Zeltner, 2020*; *Winbo et al., 2020*; *Goldsteen et al., 2022*), which apply activation of Wnt and/or SHH signaling. Nonetheless, neurons expressing either sympathetic or parasympathetic neuron markers were generated depending on BDNF and ciliary neurotrophic factor (CNTF) concentrations, as well as cell density. However, differentiation efficiencies of the optimized sympathetic- and parasympathetic-specific protocols were not reported and the efficiency of autonomic neuron induction in the combined autonomic protocol was <10%. This suggests that this approach results in high proportions of contaminating cells.

Both Wu et al. and Takayama et al. used GDNF, among other factors, for neuronal maturation (*Takayama et al., 2020*; *Wu et al., 2024b*). Once established near target tissues, parasympathetic precursors of several ganglia are initially dependent on Wnt and GDNF for proliferation (*Enomoto et al., 2000*; *Muñoz-Bravo et al., 2013*; *Knosp et al., 2015*; *Figure 6C*). The only protocol not to apply GDNF was developed by Goldsteen et al. Instead, they relied on BDNF for development and maturation of parasympathetic neurons, based on its requirement for the innervation of distal airway smooth muscle (*Radzikinas et al., 2011*). However, postganglionic parasympathetic neurons barely innervate the distal airways (*Aven and Ai, 2013*), and BDNF deletion therefore probably affects the extrinsic sympathetic, sensory, and/or vagal innervation of the lungs.

Both Takayama et al. and Wu et al. applied CNTF for neuronal maturation (*Takayama et al., 2020*; *Wu et al., 2024b*). Additionally, Takayama et al. applied NGF and NTF3 (*Takayama et al., 2020*).

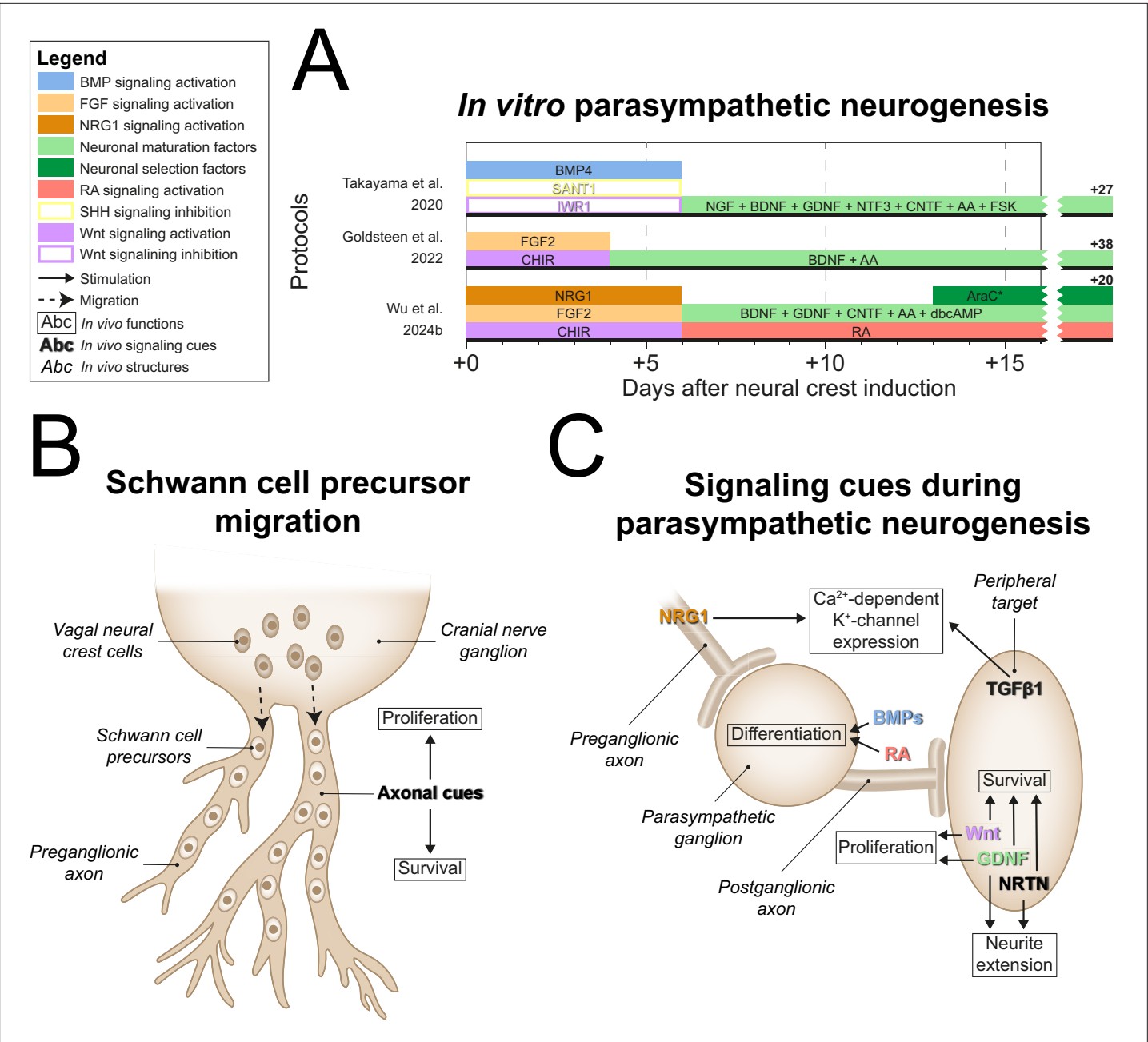

**Figure 6.** Parasympathetic neurogenesis. (**A**) Timings and signaling cues used from neural crest induction until the end of parasympathetic neuron differentiation. Duration of this phase per protocol is indicated by the horizontal black bars. Total duration of this phase per protocol is indicated to the right of the graph. Molecules targeting similar pathways have been grouped by color. (**B**) Migration of vagal neural crest-derived Schwann cell precursors along a cranial nerve. (**C**) Schematic view of the signaling requirements for parasympathetic precursor proliferation and target innervation. Signaling requirements are indicated by bold text. Signaling cues targeted by the protocols in (**A**) are indicated with colored text matching those in the figure legend. * Cytosine arabinoside selection yields optimal cell purity. However, this is not required. AA, ascorbic acid; AraC, cytosine arabinoside; BDNF, brain-derived neurotrophic factor; BMP, bone morphogenetic protein; CHIR, CHIR99021; CNTF, ciliary neurotrophic factor; dbcAMP, dibutyryl cyclic adenosine monophosphate; FGF, fibroblast growth factor; FSK, forskolin; GDNF, glial cell line-derived neurotrophic factor; NGF, nerve growth factor; NRG1, neuregulin 1; NRTN, neurturin; NTF3, neurotrophin 3; RA, retinoic acid; SHH, Sonic hedgehog; TGFβ, transforming growth factor beta.

The online version of this article includes the following figure supplement(s) for figure 6:

**Figure supplement 1.** Total overview of all parasympathetic protocols.

Although CNTF and NGF promote parasympathetic neuron survival in vitro (*Collins and Dawson, 1983*; *Adler et al., 1979*; *Tom et al., 1998*), it is at best unknown if these factors are required during embryonic development (*Stöckli et al., 1989*). Finally, Wu et al. developed the only protocol to date to apply late RA exposure (*Wu et al., 2024b*). Knockdown of RA receptor β in chicken ciliary ganglia delays mature neurotransmitter profiles and programmed cell death, characteristic of neuronal maturation (*Koszinowski et al., 2015*).

Upon maturation, parasympathetic neurons generally switch dependency from GDNF to NRTN (*Enomoto et al., 2000*; *Laurikainen et al., 2000*; *Heuckeroth et al., 1999*). Likewise, parasympathetic intrinsic airway neurons rely on GDNF family ligands for survival, one of which is likely NRTN (*Langsdorf et al., 2011*). Nonetheless, NRTN was not included in any parasympathetic protocol, and we believe this would be a promising candidate to improve parasympathetic neuron maturity.

## Which molecular definitions of autonomic neurons are applied in vitro?

Whereas neurons can be classified on the basis of their anatomical location in vivo, the demonstration of specific protein or gene expression is required to distinguish the many different types of neurons that can be generated from hPSCs in vitro. Therefore, the various markers used to define sympathetic neurons in the articles included in this review were collected and ordered by the number of articles each marker was used in (*Figure 7A and B*). If different definitions were applied within an article per technique, priority was given to the definitions used for quantification as a proportion of cells generated (i.e., flow cytometry or quantitative immunofluorescence microscopy), followed by qualitative immunofluorescence microscopy, and quantitative reverse transcriptase polymerase chain reaction (RT-qPCR). Descriptions and expression patterns of each marker are provided in *Supplementary file 5*.

## Sympathetic neuron definitions

As expected, the catecholaminergic enzymes were popular markers; TH and/or dopamine β hydroxylase (DBH) featured in 83% (24/29) of sympathetic neuron definitions. TH is essential for the synthesis of all catecholamines, whereas DBH is only required for noradrenaline and adrenaline synthesis. The (nor)adrenergic transcription factors achaete-scute family bHLH transcription factor 1 (ASCL1), paired-like homeobox 2B (PHOX2B), or GATA binding protein 3 (GATA3) also featured in 38% (11/29) of sympathetic neuron definitions. The most frequently combined markers were TH and peripherin (PRPH) (48%, 14/29), an intermediate filament mostly expressed in the cytoskeleton of neurons with axons projecting outside the CNS (*Romano et al., 2022*).

Considering that the combined expression of TH and PRPH is likely limited to catecholaminergic cells outside the CNS in vivo, we recommend demonstrating the expression of this combination, or a similar combination indicative of peripheral and catecholaminergic identity, to confirm sympathetic neuron identity. The expression of PRPH also likely excludes other peripheral NCC-derived catecholaminergic cells, such as chromaffin cells and small intensely fluorescent cells, although both cell types can also be distinguished from sympathetic neurons on a morphological basis (*Takaki et al., 2015*; *Derer et al., 1989*; *Troy et al., 1990*; *Ahonen, 1991*). The largest caveat to defining sympathetic neurons by expression of TH and PRPH is the potential confusion with immature parasympathetic and enteric neurons. Both neuron subtypes express PRPH from early development throughout adulthood, and express TH transiently during embryonic development (*Landis et al., 1987*; *Teitelman et al., 1979*; *Supplementary file 5*). Therefore, ideally new sympathetic differentiation protocols should demonstrate low expression of enteric or parasympathetic neuron markers, like choline O-acetyltransferase (CHAT), in mature sympathetic neurons.

## Parasympathetic neuron definitions

All six parasympathetic articles partly based their parasympathetic neuron definitions on the cholinergic enzyme CHAT (*Takayama et al., 2020*; *Takayama et al., 2023*; *Akagi et al., 2024a*; *Akagi et al., 2024b*; *Goldsteen et al., 2022*; *Wu et al., 2024b*). However, several other peripheral neurons also express CHAT, of which enteric neurons and cholinergic sympathetic neurons are derived from NCCs as well. This highlights the need for additional markers to support specific parasympathetic neuron identity. Other markers combined with CHAT in parasympathetic neuron articles included PHOX2B (*Takayama et al., 2020*; *Takayama et al., 2023*; *Akagi et al., 2024b*), PRPH (*Akagi et al.,*

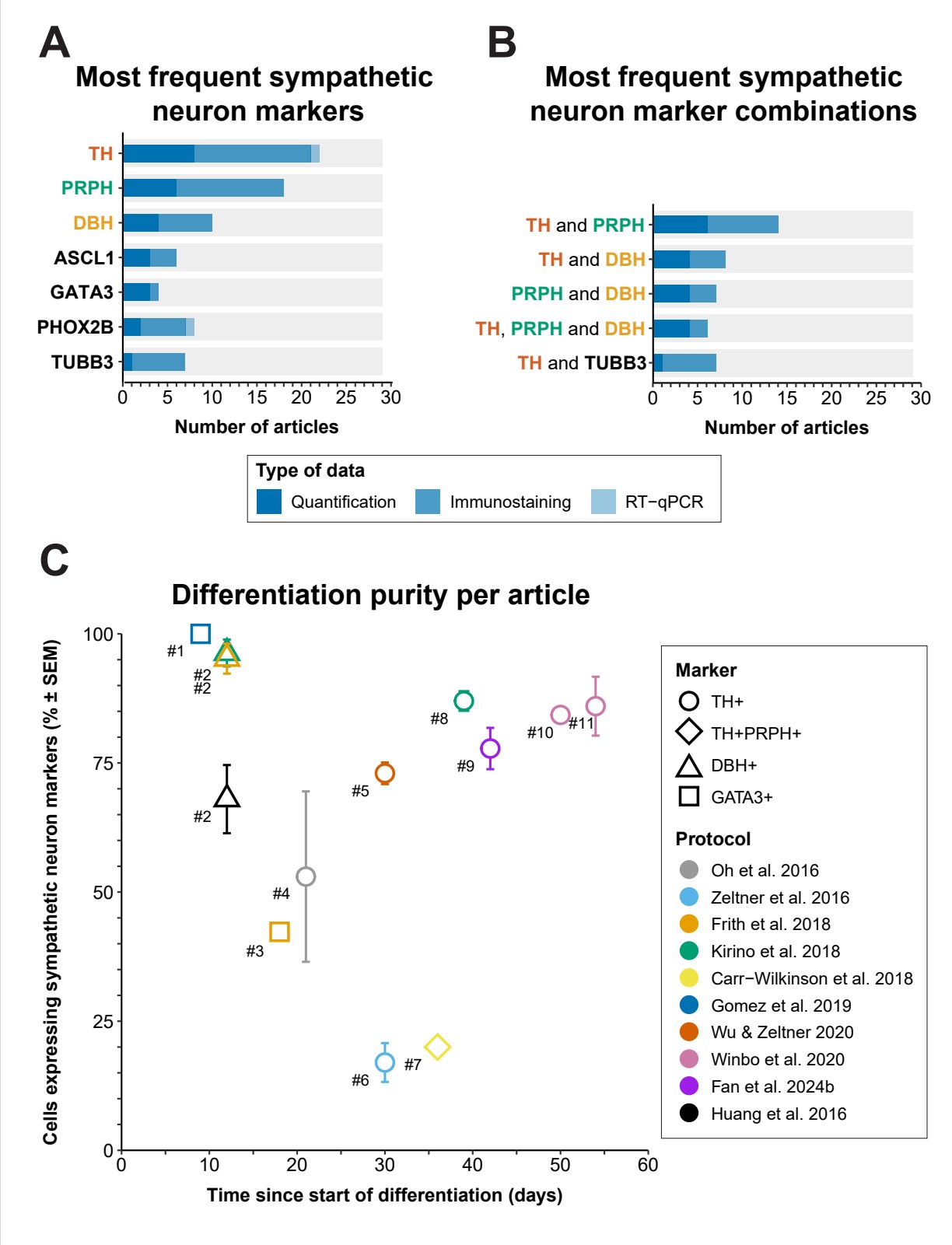

**Figure 7.** Sympathetic neuron definitions and differentiation efficiency. (**A**) All sympathetic neuron markers used in ≥3 articles, stratified by technique. (**B**) All combinations of sympathetic neuron markers used in ≥6 articles. Markers featured in multiple combinations are marked by colored text in (**A**) and (**B**). (**C**) Scatter plot of protocol purity and time of quantification per article. The graph shows only the latest timepoint per article %TH+ (or %GATA3 + or %DBH+, if %TH+ was not determined) was measured. Shapes indicate the markers used for quantification and protocol applied per article is indicated

*Figure 7 continued on next page*

*Figure 7 continued*

by color. #1 *Gomez et al., 2019*, #2 *Cheng et al., 2024*, #3 *Frith et al., 2018*, #4 *Oh et al., 2016*, #5 *Wu et al., 2022b*, #6 *Zeltner et al., 2016*, #7 *Carr-Wilkinson et al., 2018*, #8 *Kirino et al., 2018*, #9 *Fan et al., 2024a*, #10 *Li et al., 2023*, #11 *Winbo et al., 2020*. Sample sizes per article can be found in *Source data 1*. ASCL1, achaete-scute family bHLH transcription factor 1; DBH, dopamine beta-hydroxylase; GATA3, GATA binding protein 3; RT-qPCR, quantitative reverse transcriptase polymerase chain reaction; PHOX2B, paired-like homeobox 2B; PRPH, peripherin; SEM, standard error of the mean; TH, tyrosine hydroxylase; TUBB3, tubulin beta 3 class III.

*2024a*; *Akagi et al., 2024b*; *Wu et al., 2024b*), and the pan-neuronal marker, tubulin β3 (TUBB3) (*Goldsteen et al., 2022*).

However, all these markers are also expressed by cholinergic enteric and cholinergic sympathetic neurons (*Espinosa-Medina et al., 2014*; *Furlan et al., 2016*; *Tiveron et al., 1996*; *Supplementary file 5*). To exclude cholinergic sympathetic neuron identity, we recommend demonstrating the expression of H6 family homeobox 2 (HMX2) or H6 family homeobox 3 (HMX3) (*Ernsberger et al., 2020*). Although HMX2 and HMX3 were not included in their parasympathetic neuron quantification, Wu et al. demonstrated expression of both these markers in their parasympathetic neurons (*Wu et al., 2024b*). However, at least *Hmx3* is also expressed in the enteric nervous system (*Heanue and Pachnis, 2006*). Moreover, some enteric neurons also stem from vagal NCCs and SCPs, thereby overlapping the developmental path of parasympathetic neurons (*Espinosa-Medina et al., 2017*). Until single-cell RNA sequencing data of peripheral cholinergic neurons is available to identify specific non-anatomical markers of identity, we believe it will be difficult to distinguish PSC-derived parasympathetic neurons and cholinergic enteric neurons in vitro.

## How efficient are the current differentiation strategies to generate autonomic neurons?

To assess the practical applicability of each protocol, we focused on the percentage of total cells expressing autonomic neuron markers, and differentiation time. Eleven articles, applying a total of 10 sympathetic protocols, quantified sympathetic neuron purity (*Cheng et al., 2024*; *Oh et al., 2016*; *Zeltner et al., 2016*; *Frith et al., 2018*; *Kirino et al., 2018*; *Carr-Wilkinson et al., 2018*; *Gomez et al., 2019*; *Wu et al., 2022b*; *Winbo et al., 2020*; *Li et al., 2023*; *Figure 7C*). If the outcome was available, we reported %TH-positive cells or %TH- and PRPH-positive cells, or if unavailable, the percentage of cells positive for DBH, or GATA3, a sympathetic neuron marker downstream to PHOX2B (*Martik and Bronner, 2017*), was reported.

Gomez et al. were the earliest to quantify the percentage of cells expressing autonomic neuron markers, at 9 days of differentiation, and were the only ones to report GATA3 expression in 100% of the cells (*Gomez et al., 2019*). However, this should be interpreted cautiously; only 50% of cells expressed ASCL1. In addition, GATA3 is widely expressed during development, including expression in the non-neural ectoderm (*Home et al., 2009*; *Sheng and Stern, 1999*). Moreover, as was the case for the cells reported by *Cheng et al., 2024*, the cells did not show neuronal morphology at this stage.

Generally, the acceptable minimal differentiation purity will differ per research goal. However, as an arbitrary rule of thumb, we recommend that at least two-thirds of cells in culture express relevant autonomic neuron markers. The only protocols to achieve ≥67% sympathetic neuron marker expression in cultures with neuronal morphology were those described by *Kirino et al., 2018*, *Winbo et al., 2020*; *Li et al., 2023*, *Wu et al., 2022b*, and *Fan et al., 2024a*. However, all these protocols took ≥30 days to complete, which necessitates considerable time and material costs compared to more rapid protocols. To increase the number of cells generated per round of differentiation, three of these protocols featured an optional expansion phase of precursor populations for up to 2 weeks or longer before terminal differentiation (*Kirino et al., 2018*; *Wu et al., 2022b*; *Fan et al., 2024a*).

Generally, little data was available on parasympathetic differentiation efficiency. Of the two reported quantifications, only the protocol published by Wu et al. attained a ≥67% efficient parasympathetic neuron purity with 81.5 ± 2.0% CHAT-positive cells on day 30 of differentiation using cytosine arabinoside (AraC) selection (*Wu et al., 2024b*). The other reported quantification of the protocol by Goldsteen et al. showed much less pure parasympathetic neuron populations, with a mean of 27.7 ± 3.8% CHAT- and TUBB3-positive cells on day 50 of differentiation (*Goldsteen et al., 2022*).

## Which functional characteristics are shown by autonomic neurons generated in vitro?

Many disease modeling applications of autonomic neurons require neurons that are capable of functional neuron firing and interactions with other cell types. To model in vivo neuronal functions, autonomic neurons should synthesize appropriate neurotransmitters, generate action potentials in response to nicotinic stimuli, and form synapses capable of functionally influencing target cell types. These neuronal functions have been demonstrated by a majority of autonomic protocols (11/17) (*Oh et al., 2016*; *Zeltner et al., 2016*; *Frith et al., 2018*; *Kirino et al., 2018*; *Wu and Zeltner, 2020*; *Winbo et al., 2020*; *Fan et al., 2024a*; *Takayama et al., 2020*; *Goldsteen et al., 2022*; *Wu et al., 2024b*) with varying degrees of success (*Figure 2—figure supplement 1*).

## Neurotransmitter synthesis

Neurotransmitter synthesis is essential to neuron function. Moreover, determining the presence of noradrenaline or acetylcholine provides specific information on neuronal identity. This was frequently measured (11/17) (*Oh et al., 2016*; *Zeltner et al., 2016*; *Frith et al., 2018*; *Kirino et al., 2018*; *Wu and Zeltner, 2020*; *Winbo et al., 2020*; *Fan et al., 2024a*; *Takayama et al., 2020*; *Goldsteen et al., 2022*; *Wu et al., 2024b*) either via ELISA or high-performance liquid chromatography. Eight protocols successfully demonstrated the presence of appropriate neurotransmitters in culture medium (*Oh et al., 2016*; *Frith et al., 2018*; *Kirino et al., 2018*; *Wu and Zeltner, 2020*; *Winbo et al., 2020*; *Fan et al., 2024a*; *Goldsteen et al., 2022*; *Wu et al., 2024b*). However, only three protocols measured neurotransmitter concentrations in culture medium following spontaneous release (*Wu et al., 2022b*; *Fan et al., 2024a*; *Wu et al., 2024b*). All others relied on non-physiological stimulatory cues like potassium chloride or optogenetic stimulation for neurotransmitter release. By instead using nicotine as a stimulatory cue, future protocols could simultaneously provide evidence for functional nicotinic acetylcholine receptors, a feature of all autonomic neurons (*Wehrwein et al., 2016*). Furthermore, determining the presence of additional neurotransmitters such as neuropeptide Y might further specify the identity of PSC-derived autonomic neurons (*Ernsberger et al., 2020*).

## Electrophysiology

Crucial to neuronal function, neurons integrate and pass on signals through action potential generation. Nine autonomic protocols provided evidence of action potential generation, via multi-electrode array (MEA) recordings, cytosolic [$Ca^{2+}$] imaging, or whole-cell patch clamp recordings (*Oh et al., 2016*; *Frith et al., 2018*; *Wu and Zeltner, 2020*; *Winbo et al., 2020*; *Fan et al., 2024a*; *Takayama et al., 2020*; *Goldsteen et al., 2022*; *Wu et al., 2024b*). MEAs can record neuron firing rates of large

**Table 2.** Patch clamp recordings of hPSC-derived sympathetic neurons.

Electrophysiological characteristics of hPSC-derived sympathetic neurons determined by whole-cell patch clamp. Data from primary adult murine thoracic sympathetic neurons is included for reference. Tabulation is in chronological order. Data is reported as mean ± SEM or range, unless indicated otherwise. AP, action potential; hPSC, human pluripotent stem cell; NR, not reported; SEM, standard error of the mean.

| | Adult murine thoracic sympathetic neurons (McKinnon et al., 2019) (n=35) | Oh et al., 2016 (n=9) | Frith et al., 2018 (n=14) | Winbo et al., 2020 (n=30) | Takayama et al., 2023 (n=113) |
|---|---|---|---|---|---|
| Age (days) | 37–379 (postnatal) | 28 | >20 | 48–76 | >42 |
| Membrane capacitance (pF) | 89 ± 4.6 (n=34) | NR | 11 ± 0.6 | 85 ± 5.1 | NR |
| Current injection range (pA) | 0–200 | 0–800 | –10 to 100 | 0–300 | –100 to 300 |
| Proportion neurons firing repetitive APs, % | 100 | 56 | 21 | 73 | 36 |
| Resting membrane potential (mV) | –60 ± 1.1 | –46 ± 5.4 | –54 to –60 | –60 ± 1.9 | NR |
| AP amplitude (mV) | 54 ± 2.7 | NR | NR | 93 ± 3.9 | 74 ± 4.3 (n=20) |
| AP duration, half-width (ms) | 4.6 ± 0.2 | NR | NR | 2.8 ± 0.2 | NR |

areas of neurons over time or in response to stimuli, but provide limited information on the specific electrophysiological characteristics of individual neurons. In total, MEA recordings were reported for four protocols, revealing spontaneous firing in all cases (*Wu et al., 2022a*; *Takayama et al., 2023*; *Goldsteen et al., 2022*; *Wu et al., 2024b*).

To measure electrophysiological characteristics and the associated action potential dynamics of individual neurons, cytosolic [$Ca^{2+}$] imaging or patch clamp can be used. Cytosolic [$Ca^{2+}$] imaging visualizes the calcium transients associated with electrical activity, but does not directly measure voltages and currents (*Ali and Kwan, 2020*). The most direct method to measure individual neuron electrophysiology is by whole-cell patch clamp recordings, reported for four protocols (*Table 2*; *Oh et al., 2016*; *Frith et al., 2018*; *Winbo et al., 2020*; *Fan et al., 2024a*). For lack of any primary human sympathetic neuron patch clamp data, whole-cell patch clamp data of adult murine thoracic sympathetic neurons have been added to *Table 2* for reference (*McKinnon et al., 2019*). All four protocols demonstrated in- and outward voltage-sensitive currents, and *Winbo et al., 2020*, *Winbo et al., 2021* and *Fan et al., 2024b* also demonstrated spontaneous firing. Although all adult murine sympathetic neurons fire repetitively (i.e., show tonic activation) following current injection, a substantial portion of the hPSC-derived sympathetic neurons produce only single action potentials (i.e., display phasic activation) after current injection.

Altogether, the electrophysiological characteristics measured by Winbo et al. most closely resemble those of adult murine sympathetic neurons. However, this may be due to the longer culture time of these neurons compared to the other protocols. When Winbo et al. measured sympathetic neurons after only 28–41 days of differentiation, resting membrane potentials were significantly less polarized and action potential kinetics significantly slower than at 48–76 days (*Winbo et al., 2020*). Wu et al. also showed increased spontaneous firing rates with extended culture times (*Wu et al., 2022b*). Together, this emphasizes the importance of prolonged differentiation time for electrophysiological maturation.

## Functional interactions with other cell types

The ultimate result of autonomic neuronal function in vivo is the establishment of functional changes in target cells. The rapid changes in cardiomyocyte beating rates following autonomic neuronal firing represent a practical way to demonstrate target cell coupling. In total, five sympathetic (*Oh et al., 2016*; *Wu et al., 2022b*; *Winbo et al., 2020*; *Fan et al., 2024b*; *Takayama et al., 2020*) and two parasympathetic protocols (*Takayama et al., 2020*; *Wu et al., 2024b*) showed altered beating rates of cardiomyocytes in co-culture with autonomic neurons following nicotinic stimulation. Although both studies applying parasympathetic neurons adequately accounted for this, readers should note that nicotine administration can cause subtle decreases of hPSC-derived cardiomyocyte spontaneous beating rates, even in the absence of co-cultured autonomic neurons (*Winbo et al., 2020*; *Fan et al., 2024b*; *Takayama et al., 2020*). Therefore, we recommend including a cardiomyocyte monoculture control for nicotine reactivity experiments.

Besides cardiomyocytes, other cell types have also successfully formed functional interactions with hPSC-derived autonomic neurons. Wu et al. also showed hPSC-derived parasympathetic neurons could increase calcium flux in salivary acinar cells upon nicotine administration (*Wu et al., 2024b*). Finally, autonomic neurons generated by Wu et al. and Fan et al. demonstrated specific interactions with adipocyte-like cells (*Fan et al., 2024a*; *Wu et al., 2024b*). Co-culture with hPSC-derived sympathetic neurons caused human adipose-derived stem cells to increase lipid hydrolysis and adopt brown-like adipocyte identities (*Fan et al., 2024a*). Conversely, co-culture with hPSC-derived parasympathetic neurons caused 3T3-L1-derived mouse adipocytes to adopt mature morphology and increase adipogenesis (*Wu et al., 2024b*).

All-in-all, we recommend that new autonomic protocols show nicotine-dependent neuron firing or nicotine-dependent functional interactions with target cells to prove neuron functionality. Both these approaches demonstrate neuronal function from the activation of postsynaptic nicotinic receptors to the generation of an action potential. Ideally, action potential kinetics should also be measured to provide information on neuron identity and maturity.

## Discussion

In this systematic review, we have shown that most current methods to generate human autonomic neurons are hPSC-based and aimed at sympathetic neuronal differentiation. In contrast to their

parasympathetic counterparts, hPSC-derived sympathetic neurons have already been applied in multiple studies modeling the sympathetic contribution to diseases including familial dysautonomia (*Zeltner et al., 2016*; *Wu et al., 2022b*), congenital central hypoventilation syndrome (*Amer-Sarsour et al., 2024*), long QT syndrome (*Winbo et al., 2021*), Parkinson's disease (*Saleh et al., 2024*), and diabetic autonomic neuropathy (*Wu et al., 2023*). Our comparison between hPSC-derived autonomic neuronal differentiation protocols and the embryonic development of the ANS has highlighted a number of unexplored in vitro signaling cues for parasympathetic, but also for sympathetic neuronal differentiation. Additionally, we provided an overview and outlined the challenges of molecular strategies to define hPSC-derived autonomic neurons in vitro.

## Comparison to embryonic development

Most sympathetic protocols had a firm basis in embryology. Nonetheless, a number of signals vital to sympathetic neuron development in vivo were absent from all protocols, such as SEMA3A, required for the patterning of the sympathetic trunk and target innervation (*Kawasaki et al., 2002*; *Ieda et al., 2007*). Other factors involved in the development of sympathetic neurons, such as NRG1, CXCL12, and artemin, were omitted from all sympathetic protocols. The common denominator for these factors is that they primarily regulate cell migration or axon extension. These functions are probably less crucial in vitro, where cells do not require migration or axon extension to encounter signaling cues. However, factors like artemin, NRTN, endothelins, and netrin 1 may be required to recapitulate the target-specific phenotypical sympathetic neuron diversity observed in vivo (*Furlan et al., 2016*). For parasympathetic neurons, besides NRTN, TGFβ1 is another promising candidate to improve neuron maturation. In vivo, TGFβ1 regulates the expression of $Ca^{2+}$-activated $K^+$-channels together with NRG1 (*Cameron et al., 2001*; *Cameron et al., 1998*).

Embryological studies often provide inspiration for in vitro differentiation strategies, but the reverse can also be true. Although few experiments were specifically designed to answer embryological questions via the differentiation of hPSCs, the ability to induce sympathetic neuron markers in hPSCs with signaling cues observed in murine and avian models suggests that human sympathetic developmental signaling cues in vivo have not diverged notably from other amniotes. From this perspective, only the prominent role of SB-mediated SMAD2/3 inhibition in hPSC-derived NCC induction in this review raises questions about the role of this pathway in vivo. Other outcomes of embryological interest are the other cell types generated by autonomic neuron differentiation protocols. Animal models have shown that glial cells, sensory neurons, enteric neurons, and chromaffin cells, among others, are closely related to autonomic neurons (*Martik and Bronner, 2017*). The other cell types generated by the protocols in this review support the idea that these cell types and autonomic neurons are also derived from common progenitors in humans. In the protocol developed by Zeltner et al., which is one of the few protocols without BMP, contaminating cell types were shown to express α-smooth muscle actin (myofibroblast marker), or brain-specific homeobox/POU domain protein 3A (*POU4F1*, also known as *BRN3A*; sensory neuron marker) (*Zeltner et al., 2016*).

## In vitro autonomic neuron definitions

Besides markers expressed by the mature neurons, autonomic neuron identity was usually supported during differentiation by demonstrating the intermediate presence of NCC markers NGFR, HNK1, and/or SOX10 (or CD49d, which correlates with SOX10 expression; *Fattahi et al., 2016*). Of these, SOX10 seems most robustly expressed in both human premigratory and migratory human NCCs (*Betters et al., 2010*), but all markers are also expressed in parts of the neural tube, underscoring the need for multiple NCC markers in vitro.

After NCC induction, SCP formation is another specific feature of parasympathetic neuron development. However, only one article demonstrated the intermediate presence of SCPs by SOX10 expression and Schwann cell differentiation potential (*Wu et al., 2024b*). Considering that NCCs also express SOX10 (*Betters et al., 2010*), markers such as myelin protein 0 and cadherin 19 should be used to distinguish SCPs from NCCs (*Solovieva and Bronner, 2021*).

Even after establishing intermediate SCP identity, parasympathetic neurons and cholinergic enteric neurons cannot be distinguished from one another. The field currently lacks a comprehensive transcriptomic characterization of parasympathetic neurons (*Ernsberger et al., 2020*), as exists for sympathetic neurons (*Furlan et al., 2016*), to address this issue. A recent investigation of the right atrial

ganglionic plexus, which contains parasympathetic neurons, may provide some clues (*Moss et al., 2021*). However, the neurons of the intrinsic cardiac ganglia are diverse (*Fedele and Brand, 2020*). Ideally, single-cell RNA sequencing of peripheral cholinergic neurons would reveal specific molecular parasympathetic markers for future PSC-derived parasympathetic protocols.

## Protocol efficiency

Quantitative efficiency parameters largely determine the practical applicability of a particular differentiation protocol. Several sympathetic protocols have reported ≥67% TH-positive cells, but only one parasympathetic protocol achieved ≥67% CHAT-positive cells. Furthermore, although outpacing the in vivo rate of differentiation (*Kruepunga et al., 2021*), differentiation durations of protocols with high purity (≥30 days) remain a significant barrier to widespread protocol implementation. A final important metric in this category, especially for industrial applications, is cost-effectiveness. In this regard, future protocol development should include attempts to replace growth factors with less expensive small molecules (*Zhu et al., 2010*). Forced expression of lineage-specifying transcription factors like neurogenin 2 may be another way to reduce costs by decreasing differentiation time (*Hulme et al., 2022*). The same is likely to be true for autonomic neuronal cell lines derived by conditional immortalization.

## Neuronal maturation

From a functional point of view, a major challenge for hPSC-derived neurons is to develop from electrically passive hPSCs to mature neurons within a short time span. Besides prolonged time in culture, other strategies to improve autonomic neuronal maturity include the generation of microtissues containing multiple cell types. As an example, addition of hPSC-derived cardiac fibroblasts to three-dimensional (3D) microtissues consisting of hPSC-CMs and cardiac endothelial cells has been shown to improve hPSC-CM maturity (*Giacomelli et al., 2020*). Co-culture with epicardium-derived cells has also been shown to improve cardiomyocyte maturation and to stimulate sympathetic ganglion neurite outgrowth (*Weeke-Klimp et al., 2010*; *Ge et al., 2020*). Satellite glial cells, which support neuronal survival and firing in vivo (*Hanani and Spray, 2020*), are other interesting candidates to assist autonomic neuronal maturation.

## Three-dimensional autonomic neuronal models

In addition to glial cells, autonomic neurons neighbor several other cell types in vivo, such as pericytes and endoneurial fibroblasts (*Mapps et al., 2022*). Exposure to the signals and extracellular matrix deposited by these cells can influence culture stability, as well as gene and protein expression levels (*Chaicharoenaudomrung et al., 2019*). Ultimately, 3D cultures composed of multiple cell types could potentially capture disease phenotypes that two-dimensional (2D) cultures consisting mainly of neurons may not be accurate or sensitive enough for. However, although several articles report the self-assembly of autonomic neurons into ganglia-like clusters (*Frith et al., 2018*; *Wu and Zeltner, 2020*; *Takayama et al., 2020*), all current protocols have been developed for 2D culture. Considering the variety of cells that is likely to impact autonomic neuron function, and the 3D anatomy of autonomic ganglia in vivo, the field would benefit from the availability of 3D models of autonomic ganglia with multiple cell types.

## Conditional immortalization

A final lacuna revealed by our search was a complete lack of lines of immortalized human autonomic neurons. Immortalization strategies promise low proportions of contaminating cell types and fast neuron generation. The cell line probably closest to fitting our definitions would be the catecholaminergic SH-SY5Y cell line. Not strictly sympathetic, this line was derived from a human neuroblastoma metastasis. Despite its ability to express DBH in addition to TH, SH-SY5Y has widely been used to model dopaminergic neurons (which do not express DBH) to research Parkinson's disease (*Xicoy et al., 2017*). However, among other drawbacks, expression profiles of terminally differentiated SH-SY5Y-derived neurons appear inconsistent between different studies (*Bell and Zempel, 2022*), limiting their ability to model specific neuronal subtypes. This limitation could be circumvented by direct immortalization of specific neuronal subtypes. Moreover, the possibility of switching proliferation off after immortalization (i.e., conditional immortalization) could aid the recovery of mature

features upon differentiation, as has been shown for human atrial myocytes (*Harlaar et al., 2022*), another cell type with limited mitotic capacity.

## Limitations

Our comparison of in vitro differentiation methods to embryonic development has a few limitations. Although we focused mainly on the signaling molecules authors explicitly reported, as these are likely to have been selected based on experimental evidence, other growth factors were also often present (*Supplementary file 3*). For instance, although at first glance only Frith et al. include FGF2 from the start of differentiation (*Frith et al., 2018*), other protocols initiate differentiation in the PSC maintenance media StemPro, which contains FGF2 and IGF1, among others (*Akopian et al., 2010*), or StemFlex, which likely also contains FGF2. Other media contained proprietary supplements, like knockout serum replacement, of which the exact composition is not provided (*Price and Tilkins, 1998*). Moreover, multiple protocols contained undefined components like the factors secreted by PA6 cells, or fetal bovine serum, bovine serum albumin, and Matrigel, which contain varying quantities of undetermined growth factors. Finally, no studies measured or defined the levels of endogenous signaling molecules secreted by cells during differentiation. For example, spontaneously differentiating human embryonic stem cells (hESCs) produce endogenous BMP2 (*Pera et al., 2004*) and endogenous Wnt signaling varies between hESC clones (*Blauwkamp et al., 2012*). The approach closest to addressing endogenous variation was the 'top-down BMP inhibition' (*Hackland et al., 2017*) applied by two protocols (*Frith and Tsakiridis, 2019*; *Hackland et al., 2019*), which combines BMP inhibition with BMP stimulation to limit the influence of endogenous signals. Together, these factors obscure a total overview of active signaling cues in autonomic neuron protocols.

Furthermore, the scope of this review was limited to human autonomic neurons derived from hPSCs or via immortalization of primary human cells. However, alternative strategies could not be retrieved by our search strategy, such as direct reprogramming. For example, although murine, noradrenergic neurons and ganglion organoids containing multiple autonomic neuron types have been generated by overexpression of combinations of catecholaminergic transcription factors in astrocytes or mouse embryonic fibroblasts (*Li et al., 2019*; *Liu et al., 2023*). This may present another feasible route to generate human sympathetic-like neurons.

## Conclusions

The derivation of sympathetic neurons from hPSCs has progressed tremendously since the first attempts in 2016, mostly mirroring signaling cues observed in vivo. These protocols have now reached a phase in which they can be applied to human disease modeling, as evidenced by various articles included in this review. Future sympathetic protocols should consider the generation of subtypes of sympathetic neurons by incorporating target-specific factors. By comparison, the generation and application of hPSC-derived parasympathetic neurons lag behind, with few available approaches and a lack of extensive electrophysiological characterization. A significant hurdle to generating parasympathetic neurons in vitro is a lack of established specific markers. Finally, attempts to generate human autonomic neurons through conditional immortalization have not yet been reported, but will likely complement the existing hPSC-based approaches to produce these valuable but otherwise near unobtainable cells.

## Materials and methods

This systematic review was conducted following the Preferred Reporting Items for Systematic reviews and Meta-Analyses (PRISMA) 2020 guidelines (checklist is provided as *Supplementary file 6*; *Page et al., 2021*). Accordingly, a systematic review protocol was registered with Open Science Framework prior to data collection (doi:10.17605/OSF.IO/E9VGU). The protocol was amended before data collection by widening article eligibility from cardiac or unspecified sympathetic or parasympathetic neurons, to all postganglionic sympathetic or parasympathetic neurons, after it became apparent that most protocols did not aim to generate organ-specific neuronal subpopulations. This led to the inclusion of one additional article, that is, *Goldsteen et al., 2022*.

## Search strategy and article selection

The search strategy (see Appendix 1) was compiled with support from a medical information specialist. Queries were designed to retrieve all articles including in vitro research with hPSCs or immortalization, and human autonomic neurons or autonomic precursors. No additional limits or filters were applied. The PubMed (databases: Medical Literature Analysis and Retrieval System Online, PubMed Central, and Bookshelf), Ovid (database: Embase), and Web of Science (database: Web of Science Core Collection) interfaces were last searched on November 13, 2024. During full-text screening, the reference list of each article was screened for further eligible articles.

All articles were deduplicated in EndNote 20 as previously described (*Bramer et al., 2016*). Next, two reviewers (EP and TAB) independently screened titles and abstracts for potential eligibility using Rayyan (https://www.rayyan.ai/; *Ouzzani et al., 2016*). After resolving all discrepancies via discussion, a single reviewer (TAB) assessed the selected full-text records for eligibility. In cases of uncertain eligibility, eligibility was verified by discussion with two other reviewers (MCDR and MRMJ). Finally, articles that met each of the following criteria were selected:

1. An English peer-reviewed full-text record was available.
2. The article described original experiments and/or methods.
3. Techniques involving hPSCs or immortalized cells were applied.
4. The authors aimed to produce sympathetic or parasympathetic postganglionic neurons, or neurons that strongly resemble these cell types.
5. The neurons were human.
6. The authors aimed to induce homogenous populations of sympathetic or parasympathetic neurons.
7. The article provided, or referred to, methodological details and any characterization outcomes.

## Quality assessment

At the time of publication, no widely applied guidelines to assess methodology and reporting of in vitro studies existed. Therefore, we developed criteria based on the Consolidated Standards of Reporting Trials (CONSORT) and Animal Research: Reporting of In Vivo Experiments (ARRIVE) guidelines (*Schulz et al., 2010*; *Percie du Sert et al., 2020*), supplemented with self-devised criteria specific for studies involving hPSCs and neuronal cell culture (*Supplementary file 2*). Using these criteria, two reviewers (EP and TAB) independently judged each eligible record, resolving discrepancies via discussion.

## Data collection

All data was extracted manually from eligible records by a single reviewer (TAB) in Microsoft Excel and validated by a second reviewer (EP). Publication metadata, and information about materials, conditions, timepoints, and methods for deriving neurons were collected from all eligible articles. Additionally, data on gene and protein expression, neurotransmitter secretion, electrophysiology, morphology, and co-culture outcomes were collected. If multiple iterations of a protocol were reported, data was collected from the version yielding the highest purity of sympathetic or parasympathetic neurons, or if this was not determined, the most mature markers. Differentiation duration was defined as the time between the first day conditions deviated from hPSC maintenance conditions and the first day at which end-stage cells were analyzed by any means. Markers used to define neuronal identities were collected depending on outcome availability. If quantitative data was only provided graphically, values were derived from images.

## Statistics

All statistics were performed in R (version 4.4.2). Interrater reliability was calculated via Cohen's kappa using the irr package (https://cran.r-project.org/package=irr). The heterogeneity in applied methods and measured outcomes between articles precluded any meta-analyses. Individual study averages were reported as mean ± standard error of the mean (SEM). If not reported, SEM was derived by dividing the standard deviation by the square root of the number of observations.

## Acknowledgements

We thank José Plevier for her assistance in compiling the search strategy. Additionally, we are thankful to Ronald Slagter for drawing the illustrations in *Figures 4–6*.

## Additional information

### Funding

| Funder | Grant reference number | Author |
|---|---|---|
| Nederlandse Organisatie voor Wetenschappelijk Onderzoek | 91719346 | Monique RM Jongbloed |

The funders had no role in study design, data collection and interpretation, or the decision to submit the work for publication.

### Author contributions

Thomas A Bos, Conceptualization, Data curation, Formal analysis, Investigation, Visualization, Methodology, Writing – original draft, Writing – review and editing; Elizaveta Polyakova, Validation, Investigation, Methodology, Writing – review and editing; Janine Maria van Gils, Supervision, Methodology, Writing – review and editing; Antoine AF de Vries, Marie-José Goumans, Christian Freund, Writing – review and editing; Marco C DeRuiter, Conceptualization, Supervision, Writing – review and editing; Monique RM Jongbloed, Conceptualization, Supervision, Funding acquisition, Writing – original draft, Writing – review and editing

### Author ORCIDs

Thomas A Bos ⓘ https://orcid.org/0000-0003-2837-7355
Elizaveta Polyakova ⓘ https://orcid.org/0000-0002-9615-2871
Monique RM Jongbloed ⓘ https://orcid.org/0000-0002-9132-0418

### Decision letter and Author response

Decision letter https://doi.org/10.7554/eLife.103728.sa1
Author response https://doi.org/10.7554/eLife.103728.sa2

## Additional files

### Supplementary files

Supplementary file 1. List of excluded articles. A list of all articles that were excluded during full-text screening, including the reason of exclusion. PSCs, pluripotent stem cells.

Supplementary file 2. Quality assessment criteria and definitions. Detailed criteria and definitions per judgement per criterion.

Supplementary file 3. Methodological details of included protocols. Additional culture details per original protocol, relating to *Figures 3, 5 and 6*.

Supplementary file 4. Small molecules used in autonomic neuron protocols. Modes of actions for all small molecules featured in this article.

Supplementary file 5. Molecular autonomic neuron markers. Function and expression profiles of molecular autonomic neuron markers.

Supplementary file 6. Preferred Reporting Items for Systematic Reviews and Meta-Analyses 2020 Main Checklist. Locations of all Preferred Reporting Items for Systematic Reviews and Meta-Analyses 2020 Main Checklist items.

MDAR checklist

Source data 1. Dataset relating to *Figure 1*. All title-abstract exclusion reasons per record. Please see *Supplementary file 1* for exclusion reasons during full-text screening. n/a, not available. Dataset relating to *Figure 7A and B*. Definitions of sympathetic neurons per article. Only one of quantification, immunofluorescence or RT-qPCR data was collected per article. Quantification data

was preferentially collected over immunofluorescence or RT-qPCR. Immunofluorescence data was collected preferentially over RT-qPCR. n/a, not available; RT-qPCR, quantitative reverse transcriptase polymerase chain reaction. Dataset relating to *Figure 7C*: sympathetic neuron efficiency quantification per article. Only one efficiency marker was collected per article. %TH+ was collected preferentially over the other markers. %TH+ PRPH+ was collected preferentially over %GATA3+ and %DBH+. DBH, dopamine beta-hydroxylase; GATA3, GATA binding protein 3; n/a, not available; NR, not reported; PRPH, peripherin; TH, tyrosine hydroxylase.

## Data availability

All supporting data is provided in the tables, figures and supplementary files.

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

## Appendix 1

### PubMed query

("pluripotent stem cells"[mesh] OR "pluripotent stem cell*"[tw] OR "PSC"[tw] OR "PSCs"[tw] OR "hPSC*"[tw] OR "iPSC*"[tw] OR "hiPSC*"[tw] OR "embryonic stem cell*"[tw] OR "hESC*"[tw] OR "ESC"[tw] OR "ESCs"[tw] OR "immortal*"[tw] OR "inducible proliferati*"[tw] OR "conditional proliferati*"[tw]).

AND ("ganglia, autonomic"[mesh] OR "autonomic nervous system"[mesh] OR "ANS"[tw] OR "autonomic neuron*"[tw] OR "autonomic nerve cell*"[tw] OR "autonomic-like neuron*"[tw] OR "autonomic-like nerve cell*"[tw] OR "sympathetic neuron*"[tw] OR "SNS "[tw] OR "sympathetic-like neuron*"[tw] OR "sympathetic nerve cell*"[tw] OR "sympathetic-like nerve cell*"[tw]OR "parasympathetic neuron*"[tw] OR "parasympathetic nerve cell*"[tw] OR "PNS"[tw] OR "para sympathetic neuron*"[tw] OR "para-sympathetic nerve cell*"[tw] OR "parasympathetic-like neuron*"[tw] OR "para-sympathetic-like neuron*"[tw] OR "parasympathetic-like nerve cell*"[tw] OR "para-sympathetic-like nerve cell*"[tw]OR "sympathoadrenal progenitor*"[tw]OR "sympathoadrenal precursor*"[tw] OR "schwann cell precursor*"[tw] OR "schwann cell progenitor*"[tw]).

OR (("neural crest"[mesh] OR "neural crest cell*"[tw] OR "neural crest stem cell*"[tw] OR "neural crest-like cell*"[tw] OR "neural crest progenitor cell*"[tw] OR "NC cell*"[tw] OR "NCC"[tw]) AND ("trunk"[tw] OR "posterior"[tw] OR "vagal"[tw] OR "cardiac"[tw])).

OR (("adrenergic neuron*"[tw] OR "adrenergic nerve cell*"[tw] OR "adrenergic-like neuron*"[tw] OR "adrenergic-like nerve cell*"[tw] OR "noradrenergic neuron*"[tw] OR "noradrenergic nerve cell*"[tw] OR "noradrenergic-like neuron*"[tw] OR "noradrenergic-like nerve cell*"[tw] OR "cholinergic-like nerve cell*"[tw] OR "cholinergic neuron*"[tw] OR "cholinergic-like neuron*"[tw] OR "cholinergic nerve cell*"[tw]) AND ("peripher*"[tw] OR "postganglion*"[tw] OR "post ganglion*"[tw])).

AND ("cell differentiation"[Mesh] OR "cells, cultured"[Mesh] OR "differentiat*"[tw] OR "deriv*"[tw] OR "induc*"[tw]).

NOT ("animals"[mesh] NOT "humans"[mesh]).

### Ovid query

(exp pluripotent stem cell/ OR "pluripotent stem cell*".ti,ab. OR "PSC".ti,ab. OR "PSCs".ti,ab. OR "hPSC*".ti,ab. OR "iPSC*".ti,ab. OR "hiPSC*".ti,ab. OR "embryonic stem cell*".ti,ab. OR "hESC*".ti,ab. OR "ESC".ti,ab. OR "ESCs".ti,ab. OR "immortal*".ti,ab. OR "inducible proliferati*".ti,ab. OR "conditional proliferati*".ti,ab).

AND (exp autonomic ganglion/ OR exp autonomic nervous system/ OR "ANS".ti,ab. OR "autonomic neuron*".ti,ab. OR "autonomic nerve cell*".ti,ab. OR "autonomic-like neuron*".ti,ab. OR "autonomic-like nerve cell*".ti,ab. OR "sympathetic neuron*".ti,ab. OR "SNS ".ti,ab. OR "sympathetic-like neuron*".ti,ab. OR exp sympathetic nerve cell/ OR "sympathetic nerve cell*".ti,ab. OR "sympathetic-like nerve cell*".ti,ab. OR "parasympathetic neuron*".ti,ab. OR "parasympathetic nerve cell*".ti,ab. OR "PNS".ti,ab. OR "para-sympathetic neuron*".ti,ab. OR "para-sympathetic nerve cell*".ti,ab. OR "parasympathetic-like neuron*".ti,ab. OR "para-sympathetic-like neuron*".ti,ab. OR "parasympathetic-like nerve cell*".ti,ab. OR "para-sympathetic-like nerve cell*".ti,ab).

OR "sympathoadrenal progenitor*".ti,ab. OR "sympathoadrenal precursor*".ti,ab. OR "schwann cell precursor*".ti,ab. OR "schwann cell progenitor*".ti,ab.

OR ((exp neural crest/ OR "neural crest cell*".ti,ab. OR "neural crest stem cell*".ti,ab. OR "neural crest-like cell*".ti,ab. OR "neural crest progenitor cell*".ti,ab. OR "NC cell*".ti,ab. OR "NCC".ti,ab) AND ("trunk".ti,ab. OR "posterior".ti,ab. OR "vagal".ti,ab. OR "cardiac".ti,ab)).

OR (("adrenergic neuron*".ti,ab. OR "adrenergic nerve cell*".ti,ab. OR "adrenergic-like neuron*".ti,ab. OR "adrenergic-like nerve cell*".ti,ab. OR "noradrenergic neuron*".ti,ab. OR "noradrenergic nerve cell*".ti,ab. OR "noradrenergic-like neuron*".ti,ab. OR "noradrenergic-like nerve cell*".ti,ab. OR "cholinergic-like nerve cell*".ti,ab. OR "cholinergic neuron*".ti,ab. OR "cholinergic-like neuron*".ti,ab. OR "cholinergic nerve cell*".ti,ab.) AND ("peripher*".ti,ab. OR "postganglion*".ti,ab. OR "post-ganglion*".ti,ab)).

AND (exp cell differentiation/ OR exp cell culture/ OR "differentiat*".ti,ab. OR "deriv*".ti,ab. OR "induc*".ti,ab).

NOT ((exp animal/ OR nonhuman/) NOT exp human/).

NOT (conference OR conference abstract OR conference review).pt.

NOT ("Parkinson*".ti. OR ("dopamin*".ti,ab. AND ("Parkinson*".ti,ab. OR "substantia nigra". ti,ab. OR "striat*".ti,ab.))).

## Web of Science query

TS=("pluripotent stem cell*" OR "PSC" OR "PSCs" OR "hPSC*" OR "iPSC*" OR "hiPSC*" OR "embryonic stem cell*" OR "hESC*" OR "ESC" OR "ESCs" OR "immortal*" OR "inducible proliferati*" OR "conditional proliferati*").

AND TS=("autonomic ganglion" OR "autonomic nervous system" OR "ANS" OR "autonomic neuron*" OR "autonomic nerve cell*" OR "autonomic-like neuron*" OR "autonomic-like nerve cell*" OR "sympathetic neuron*" OR "SNS " OR "sympathetic-like neuron*" OR exp sympathetic nerve cell/ OR "sympathetic nerve cell*" OR "sympathetic-like nerve cell*" OR "parasympathetic neuron*" OR "parasympathetic nerve cell*" OR "PNS" OR "para-sympathetic neuron*" OR "para-sympathetic nerve cell*" OR "parasympathetic-like neuron*" OR "para-sympathetic-like neuron*" OR "parasympathetic-like nerve cell*" OR "para-sympathetic-like nerve cell*" OR "sympathoadrenal progenitor*" OR "sympathoadrenal precursor*" OR "schwann cell precursor*" OR "schwann cell progenitor*" OR).

(("neural crest" OR "neural crest-like cell*" OR "NC cell*" OR "NCC") AND ("trunk" OR "posterior" OR "cardiac" OR "vagal")) OR (("adrenergic neuron*" OR "adrenerg ic nerve cell*" OR "adrenergic-like neuron*" OR "adrenergic like nerve cell*" OR "noradrenergic neuron*" OR "noradrenergic nerve cell*" OR "noradrenergic-like neuron*" OR "noradrenergic-like nerve cell*" OR "cholinergic-like nerve cell*" OR "cholinergic neuron*" OR "cholinergic-like neuron*" OR "cholinergic nerve cell*") AND ("peripher*" OR "postganglion*" OR "post-ganglion*")).

AND TS=("culture*" OR "differentiat*" OR "deriv*" OR "induc*").

NOT TI=(("veterinar*" OR "animal*" OR "mouse" OR "mammalian" OR "mice" OR "rodent*" OR "murine" OR "rat" OR "rats" OR "pig" OR "pigs" OR "porcine" OR "chicken*" OR "avian" OR "zebrafish" OR "piscine" OR "rabbit*" OR "leporine") NOT "human*").

