## [Editor Report]

This valuable systematic review provides substantial insights into pluripotent stem cell-derived autonomic postganglionic neuron differentiation techniques. The methodology to collect the underlying evidence is solid. This work provides a helpful resource for researchers who want to establish differentiation pluripotent stem cell into autonomic postganglionic neurons for disease modeling.

---

## [Decision Letter]

**Decision letter after peer review:**

Thank you for submitting your article "Human autonomic postganglionic neurons for disease modelling: a systematic review in light of developmental cues" for consideration by *eLife*. Your article has been reviewed by 3 peer reviewers, one of whom is a member of our Board of Reviewing Editors, and the evaluation has been overseen by Olujimi Ajijola as the Senior Editor and by a Reviewing Editor.

The reviewers have discussed the reviews with one another and the Reviewing Editor has drafted this decision to help you prepare a revised submission. Please let us know when you plan to resubmit.

The manuscript entitled "Human autonomic postganglionic neurons for disease modelling: a systematic review in light of developmental cues" is a well-researched manuscript outlining the state of the field for the differentiation of human pluripotent stem cells. This is an important topic in the field, as accessible protocols for the creation of autonomic iPSCs that recapitulate the phenotypes of human neurons and allow modeling of disease are proving to be a powerful platform for scientific and medical research.

The authors compared in vitro autonomic neuron differentiation strategies to in vivo embryonic development of the autonomic nervous system, and highlighted unexplored in vitro signaling cues for autonomic neuronal differentiation.

As human autonomic neuronal cell models emerge as tools for disease modelling, this review will be very useful for the field of research.

Essential revisions:

The authors find that the review did assess the literature systematically and the manuscript is well written in general, but several major points came up in the review:

1. A more detailed record of how the largest number of records (969 records) were excluded would be helpful. In the same manner that the exclusion of 100 articles was detailed was informative – similar information for the 969 records (if possible) would increase confidence in the selected papers.

2. On many occasions, it seems like the authors only enumerate what has been done in the different publications without giving context. The authors should use their expertise and knowledge in this field to interpret the data and sum up the evidence for the reader. Based on their expertise, the authors should give clear recommendations on e.g. readouts for quality control (e.g. purity percentage?) and methods that should be used and reported when establishing and/or publishing new differentiation protocols.

3. The authors completely neglected to address the topic of 3D cell culture vs 2D cell culture. Were all protocols performed in 2D cell cullture? This might be a major limitation to the field, especially when considering the specific anatomy of autonomic neurons (assembled in ganglia with close intimacy of other cells types such as glia and fibroblasts). Please include this in the revised manuscript.

4. Most of the Figures are highly complex and not self-explanatory. While we appreciate the amount of work that went into them and are sure the authors have a well-thought-through colour scheme, please try to make them easier to understand for the general audience. We specifically suggest including (sub)titles, captions and legends for the different colours of bars, arrows and growth factors.

---

## [Author Response]

Essential revisions:The authors find that the review did assess the literature systematically and the manuscript is well written in general, but several major points came up in the review:1. A more detailed record of how the largest number of records (969 records) were excluded would be helpful. In the same manner that the exclusion of 100 articles was detailed was informative – similar information for the 969 records (if possible) would increase confidence in the selected papers.

Thank you for the suggestion, we agree this would increase confidence in the search process. During title abstract screening we documented exclusion reasons in Rayyan (https://www.rayyan.ai/) in case of conflicts between both reviewers. We have exported these from Rayyan and added the number of records per exclusion reason to figure 1. The dataset is available in Source Data File 1.

2. On many occasions, it seems like the authors only enumerate what has been done in the different publications without giving context. The authors should use their expertise and knowledge in this field to interpret the data and sum up the evidence for the reader. Based on their expertise, the authors should give clear recommendations on e.g. readouts for quality control (e.g. purity percentage?) and methods that should be used and reported when establishing and/or publishing new differentiation protocols.

We agree that in the first version of our manuscript, our results required the reader to make their own interpretations in many places. Below we have first described the general changes that were made per section, followed by specific quotes at the end of the response to this comment. In general, we have moved essential context and interpretations from the Discussion section to the Results section, or added new interpretations as needed.

First, in our comparison between in vitro protocols and embryology we have reorganized the neural crest section to make the rationale for the four different groups we identified more explicit. Here, we have also related the modulation of signaling pathways to the efficiency achieved in protocols modulating the pathway in question, to support several recommendations we have interweaved in this section.

Although there are too few parasympathetic articles to make meaningful groups to compare, the parasympathetic embryology text was restructured in a way that we believe better highlights the strengths and weaknesses of each protocol in comparison to embryonic development.

Next, following the overview of molecular identity markers, a separate paragraph was added to both the ‘sympathetic neuron definitions’ and ‘parasympathetic neuron definitions’ sections. Herein we have explicitly formulated recommendations supported by in vivo expression data.

Concerning recommendations for protocol efficiency, the acceptable minimal purity percentage is likely to vary based on the design and purpose of an experiment. Furthermore, there is little known about the effect of non-neuronal contamination on experimental outcomes. Therefore, we do not have any empirical data to base a minimal purity percentage recommendation on. Nonetheless, to provide the readers with a guideline, we have suggested the arbitrary proportion of two thirds. We believe this is a fair minimum proportion of the total cells to assume that most cellular interactions in the culture involve the neuronal population of interest. We have also adjusted our definition of a highly efficient protocol accordingly to two thirds throughout the text.

Finally, considering the similarities in functional readouts between sympathetic and parasympathetic neurons, we have combined both sympathetic and parasympathetic articles and discussed these per neuronal function tested. By doing so, we have avoided the necessarily enumerative description of the small number of parasympathetic protocols available. Moreover, we believe this structure provides the reader with more context on the value of each technique. We have also added a recommendation for the most optimal functional readouts for new protocols.

In summary, the following changes were made throughout the Results section:

In the results on lines 187-199 a conclusion about the similarity between embryonic development and SDIA was added:

“One of the earliest approaches, applied in two sympathetic protocols^17,27^, utilized the stromal cell-derived inducing activity (SDIA) of PA6 cells, a mouse preadipocyte cell line^58.^ Although both protocols reported the expression of catecholaminergic markers, sympathetic neuron yields were low, despite the use of fluorescence-activated cell sorting (FACS) to select for NCC markers during differentiation. Moreover, the signaling cues involved in SDIA have not been fully defined, and can also induce dopaminergic neuron differentiation^59^. Together, this suggests SDIA does not specifically recapitulate the embryonic development of sympathetic neurons.”

The sections about dual SMAD inhibition and early Wnt activation have been rewritten to explain the rationale for the distinction between both groups. The current text reads as follows on lines 215-308. We have also added more context and interpretations:

“All other approaches applied activation of Wnt signaling via CHIR99021 (CHIR)-based glycogen synthesis kinase 3 inhibition to generate NCCs (Figure 3, for additional culture details see Figure 3 —figure supplement 1). The modes of action for all small molecules used in the included protocols are provided in Figure 3 —figure supplement 2. Activation of Wnt signaling via CHIR in these protocols is in line with avian NCC development, which also depends on Wnt signaling ^60^(Figure 4A). Of protocols applying CHIR, two parasympathetic protocols^42,46^ and four sympathetic protocols^18,19,35,42^ applied a technique coined ‘dual SMAD inhibition’. These protocols combined SMAD2/3 inhibition and SMAD1/5/8 inhibition from the start of differentiation, and added CHIR from day two of differentiation. Transforming growth factor β (TGFβ)-, Activin- and Nodal-specific SMAD2/3 signaling was always inhibited via anaplastic lymphoma kinase (ALK)4/5/7 receptor inhibitor SB431542 (SB)^61^, and bone morphogenetic protein (BMP)-specific SMAD1/5/8 signaling was inhibited by small molecule inhibitors LDN193189 (LDN) or dorsomorphin (DMH)^62,63^. Although little seems to be known about SMAD2/3 signaling requirements for NCC induction in vivo, SMAD1/5/8 signaling is inhibited at the start of neural crest specification, followed by activation as neural crest development progresses^60,64^. This may parallel the use of LDN or DMH. Furthermore, in the absence of CHIR, dual SMAD inhibition instead results in high proportions of neural plate marker expression^62^. This resembles the way inhibition of Wnt induces neural plate expression in avian ectodermal explants which would otherwise express neural crest markers^60^ (Figure 4A).

The addition of CHIR to dual SMAD inhibition results in ~70% of cells expressing neural crest marker SRY-box transcription factor 10 (SOX10) by day 12 of differentiation^63^. Addition of two other small molecules (DAPT, an indirect Notch signaling inhibitor^65^, and SU5402, an inhibitor of fibroblast growth factor receptor 1 (*FGFR1*) and vascular endothelial growth factor receptor 2 (VEGFR2)^66^) slightly increases the proportion of cells expressing SOX10 by day 12 of differentiation to ~80%^63^. Oh et al. later modified this approach (i.e. modified three inhibitor approach) by substituting SU5402 for PD173074^18^, another FGFR inhibitor. The mechanisms by which SU5402 or PD173074, and DAPT contribute to NCC induction are still unknown. Although Notch signaling influences fate decisions after NCC induction, any direct role of Notch in NCC induction is unclear in chicken and mice^67^. Furthermore, fibroblast growth factor (FGF) signaling inhibition by SU5402 or PD173074 seems to contradict the FGF requirement for NCC formation in vivo^68^ (Figure 4A). On the other hand, the concentration of PD173074 used, 0.2 µM, may not entirely suppress endogenous cellular FGF signaling^29^.

CHIR-mediated Wnt signaling activation from day zero of differentiation under dual SMAD inhibition conditions dramatically reduces SOX10-positive NCC induction^69^. Therefore, we consider protocols employing CHIR from the start of differentiation (i.e. ‘early Wnt activation’) to be distinct from dual SMAD inhibition protocols. Protocols in the ‘early Wnt activation’ category often inhibited SMAD1/5/8 signaling via SB^15,24,28,30,39,40,47^, but did not inhibit BMP-specific SMAD2/3 signaling. The lack of BMP modulation in most early Wnt activation protocols seemingly contradicts evidence from embryonic development^60,64^. Similar to dual SMAD inhibition protocols, almost no early Wnt activation protocols actively stimulate FGF signaling simultaneously with Wnt stimulation, both of which are required for avian NCC development^68^. Nonetheless, three of the four protocols retrieved by our query which reported high sympathetic neuron differentiation purities (≥67% of cells expressing tyrosine hydroxylase (TH)), relied on a form of early Wnt activation without active FGF stimulation for NCC induction^24,30,40^. Information on endogenous cellular Wnt, BMP and FGF signaling in these protocols would help clarify this apparent discrepancy.”

Our interpretation of the embryonic evidence for NMP contributions to NCC induction on was also emphasized on lines 309-311: “A final induction strategy applied by Frith et al.^21^, is based on neuromesodermal progenitor (NMP) induction. Although the evidence is not conclusive, NMPs may contribute to posterior NCCs in vivo (Figure 4B), as suggested by various grafting and lineage tracing studies^71-74^.”

We have emphasized a relatively unexplored differentiation route on lines 331-334: “Strikingly, cells treated with 10 µM CHIR showed vastly higher HOXC9 expression than cells treated with 3 µM CHIR. Further investigation is warranted to explore whether this relatively simple approach to generate posterior NCCs can be used to efficiently generate mature sympathetic neurons.”

Following the section about axial identity of neural crest cells, a recommendation for future sympathetic protocols was added on lines 342-344: “Altogether, considering the instructive role HOX genes have during development^85^, we recommend demonstrating posterior HOX gene expression when establishing new sympathetic protocols.”

We have related the use of BMP to protocol efficiency on lines 356-359: “In any case, three of the four sympathetic protocols reporting high differentiation efficiencies (≥67% of cells expressing TH) actively stimulated BMP signaling^24,35,40^, suggesting this is also an important component of sympathetic neuron differentiation in vitro.”

We have related the use of SHH to protocol efficiency on lines 366-368: “However, direct stimulation of SHH signaling is only included in one of the four sympathetic protocols reporting high efficiencies (≥67% TH-positive)^35^. At most, this suggests that active stimulation of SHH signaling plays an accessory role in vitro.”

The investigation of glial cell generation in future protocols striving to recapitulate embryonic development has been recommended on lines 378-381: “In order to fully recapitulate sympathetic embryonic development, we believe a protocol should be capable of generating both glial cells and neurons from the same progenitor population, depending on Notch signaling. However, generally, the presence of glial cells was not investigated in current protocols.”

The comparison to parasympathetic embryonic development has been rewritten to follow the sequence of parasympathetic neurogenesis. In this way we believe it is clearer which protocols resemble embryonic development at each stage. The section now reads as follows on lines 429-516:

Lines 429-437: “After generating NCCs, each parasympathetic protocol employed divergent tactics to generate parasympathetic precursors (Figure 6A, full-length parasympathetic protocol overviews are shown in Figure 6 —figure supplement 1). in vivo, vagal NCCs migrate along preganglionic axons to the site of the prospective parasympathetic ganglia in the form of SCPs, which depend on axonal signals like NRG1 for survival^122-124^ (Figure 6B). Goldsteen et al. and Wu et al. showed increased expression of axial markers of vagal NCCs^46,47^, homeobox B3 (HOXB3) and homeobox B5 (HOXB5). However, only Wu et al. explicitly targeted SCP generation via NRG1 in combination with CHIR and FGF2^47^. In strong support of the recapitulation of embryonic development in this protocol, Wu et al. showed that the resulting SOX10-positive SCPs were also capable of Schwann cell generation.”

Lines 438-459: “in vivo, the development of ciliary ganglion neurons, a parasympathetic neuron subtype, depends on local BMPs, possibly BMP4, BMP5, and/or BMP7^125^. The only parasympathetic protocol to actively stimulate BMP signaling after NCC induction was the protocol by Takayama et al.^42^. As mentioned before, sympathetic ganglia also rely on BMPs during development^90,92-94^. Instead of SCP generation, this protocol inhibited alternative NCC fates, to retain progenitor populations capable of both sympathetic and parasympathetic differentiation. SHH signaling stimulates enteric neuron-fated NCC proliferation^126^, and constitutively active Wnt signaling abolishes autonomic neuron differentiation in favor of sensory neuron differentiation^127^. Therefore, the inhibition of alternative NCC fates was implemented by SHH and Wnt inhibition, via SANT1 and IWR1, respectively^42^. This strategy contrasts strikingly with the majority of other autonomic protocols (9/12)^15,18,19,21,28-30,35,46^ which apply activation of Wnt and/or SHH signaling. Nonetheless, neurons expressing either sympathetic or parasympathetic neuron markers were generated depending on BDNF and ciliary neurotrophic factor (CNTF) concentrations, as well as cell density. However, differentiation efficiencies of the optimized sympathetic- and parasympathetic-specific protocols were not reported and the efficiency of autonomic neuron induction in the combined autonomic protocol was <10%. This suggests that this approach results in high proportions of contaminating cells.”

Lines 465-472: “Both Wu et al. and Takayama et al. used GDNF, among other factors, for neuronal maturation^42,47^. Once established near target tissues, parasympathetic precursors of several ganglia are initially dependent on Wnt and GDNF for proliferation^128-130^ (Figure 6C). The only protocol not to apply GDNF was developed by Goldsteen et al. Instead, they relied on BDNF for development and maturation of parasympathetic neurons, based on its requirement for the innervation of distal airway smooth muscle^131^. However, postganglionic parasympathetic neurons barely innervate the distal airways^132^, and BDNF deletion therefore probably affects the extrinsic sympathetic, sensory and/or vagal innervation of the lungs.”

Lines 473-483: “Both Takayama et al. and Wu et al. applied CNTF for neuronal maturation^42,47^. Additionally, Takayama et al. applied NGF and NTF3^42^. Although CNTF and NGF promote parasympathetic neuron survival in vitro^134-136^, it is at best unknown if these factors are required during embryonic development^137^. Finally, Wu et al. developed the only protocol to date to apply late RA exposure^47^. Knockdown of RA receptor β in chicken ciliary ganglia delays mature neurotransmitter profiles and programmed cell death, characteristic of neuronal maturation^57^.”

Lines 512-516: *“*Upon maturation, parasympathetic neurons generally switch dependency from GDNF to NRTN^128,133,138^. Likewise, parasympathetic intrinsic airway neurons rely on GDNF family ligands for survival, one of which is likely NRTN^139^. Nonetheless, NRTN was not included in any parasympathetic protocol, and we believe this would be a promising candidate to improve parasympathetic neuron maturity.”

We have summed up the parasympathetic molecular definitions as follows on lines 568-578: “All six parasympathetic articles partly based their parasympathetic neuron definitions on the cholinergic enzyme (CHAT^42-47^). However, several other peripheral neurons also express CHAT, of which enteric neurons and cholinergic sympathetic neurons are derived from NCCs as well. This highlights the need for additional markers to support specific parasympathetic neuron identity. Other markers combined with CHAT in parasympathetic neuron articles included PHOX2B^42,43,45^, PRPH^44,45,47^, and the pan-neuronal marker, tubulin β3 (TUBB3)^46^.”

We have added an (arbitrary) recommendation about minimal purity percentage in the results on lines 608-610: “Generally, the acceptable minimal differentiation purity will differ per research goal. However, as an arbitrary rule of thumb, we recommend that at least two thirds of cells in culture express relevant autonomic neuron markers.”

Sympathetic and parasympathetic articles were combined for the functional section of our results to sum up the evidence across all articles. We have also divided the results by neuronal function to provide more context to the reader. The section now reads as follows, on lines 626-739:

Lines 626-631: “Many disease modelling applications of autonomic neurons require neurons that are capable of functional neuron firing and interactions with other cell types. To model in vivo neuronal functions, autonomic neurons should synthesize appropriate neurotransmitters, generate action potentials in response to nicotinic stimuli, and form synapses capable of functionally influencing target cell types. These neuronal functions have been demonstrated by a majority of autonomic protocols (11/17)^18,19,21,24,30,35,40,42,46,47^ with varying degrees of success (Figure 2 —figure supplement 1).”

Lines 639-651: ”*Neurotransmitter synthesis*

Neurotransmitter synthesis is essential to neuron function. Moreover, determining the presence of noradrenaline or acetylcholine provides specific information on neuronal identity. This was frequently measured (11/17)^18,19,21,24,30,35,40,42,46,47^, either via enzyme-linked immunosorbent assay (ELISA) or high-performance liquid chromatography. Eight protocols successfully demonstrated the presence of appropriate neurotransmitters in culture medium^18,21,24,30,35,40,46,47^. However, only three protocols measured neurotransmitter concentrations in culture medium following spontaneous release^32,40,47^. All others relied on non-physiological stimulatory cues like potassium chloride or optogenetic stimulation for neurotransmitter release. By instead using nicotine as a stimulatory cue, future protocols could simultaneously provide evidence for functional nicotinic acetylcholine receptors, a feature of all autonomic neurons^1^. Furthermore, determining the presence of additional neurotransmitters such as neuropeptide Y might further specify the identity of PSC-derived autonomic neurons^148^.”

Lines 657-684: “*Electrophysiology*

Crucial to neuronal function, neurons integrate and pass on signals through action potential generation. Nine autonomic protocols provided evidence of action potential generation, via multi-electrode array (MEA) recordings, cytosolic [Ca^2+^] imaging, or whole-cell patch clamp recordings^18,21,30,35,40,42,46,47^. MEAs can record neuron firing rates of large areas of neurons over time or in response to stimuli, but provide limited information on the specific electrophysiological characteristics of individual neurons. In total, MEA recordings were reported for four protocols, revealing spontaneous firing in all cases^31,43,46,47^.

To measure electrophysiological characteristics and the associated action potential dynamics of individual neurons, cytosolic [Ca^2+^] imaging or patch clamp can be used. Cytosolic [Ca^2+^] imaging visualizes the calcium transients associated with electrical activity, but does not directly measure voltages and currents^153^. The most direct method to measure individual neuron electrophysiology is by whole-cell patch clamp recordings, reported for four protocols (Table 2): Oh et al., Frith et al., Winbo et al., and Fan et al.^18,21,35,40^. For lack of any primary human sympathetic neuron patch clamp data, whole-cell patch clamp data of adult murine thoracic sympathetic neurons have been added to Table 2 for reference^154^. All four protocols demonstrated in- and outward voltage-sensitive currents, and Winbo et al.^35,36^ and Fan et al.^41^ also demonstrated spontaneous firing. Although all adult murine sympathetic neurons fire repetitively (i.e. show tonic activation) following current injection, a substantial portion of the hPSC-derived sympathetic neurons produce only single action potentials (i.e. display phasic activation) after current injection.

Altogether, the electrophysiological characteristics measured by Winbo et al. most closely resemble those of adult murine sympathetic neurons. However, this may be due to the longer culture time of these neurons compared to the other protocols. When Winbo et al. measured sympathetic neurons after only 28-41 days of differentiation, resting membrane potentials were significantly less polarized and action potential kinetics significantly slower than at 48-76 days^35^. Wu et al. also showed increased spontaneous firing rates with extended culture times^32^. Together this emphasizes the importance of prolonged differentiation time for electrophysiological maturation.”

Lines 716-738: “*Functional interactions with other cell types*

The ultimate result of autonomic neuronal function in vivo, is the establishment of functional changes in target cells. The rapid changes in cardiomyocyte beating rates following autonomic neuronal firing represent a practical way to demonstrate target cell coupling. In total, five sympathetic^18,32,35,41,42^ and two parasympathetic protocols^42,47^ showed altered beating rates of cardiomyocytes in co-culture with autonomic neurons following nicotinic stimulation. Although both parasympathetic studies adequately accounted for this, readers should note that nicotine administration can cause subtle decreases of hPSC-derived cardiomyocyte spontaneous beating rates, even in the absence of co-cultured autonomic neurons^35,41,42^. Therefore we recommend including a cardiomyocyte monoculture control for nicotine reactivity experiments.

Besides cardiomyocytes, other cell types have also successfully formed functional interactions with hPSC-derived autonomic neurons. Wu et al. also showed hPSC-derived parasympathetic neurons could increase calcium flux in salivary acinar cells upon nicotine administration^47^. Finally, autonomic neurons generated by Wu et al. and Fan et al. demonstrated specific interactions with adipocyte-like cells^40,47^. Co-culture with hPSC-derived sympathetic neurons caused human adipose-derived stem cells to increase lipid hydrolysis and adopt brown-like adipocyte identities^40^. Conversely, co-culture with hPSC-derived parasympathetic neurons caused 3T3-L1-derived mouse adipocytes to adopt mature morphology and increase adipogenesis^47^.”

To avoid repeating the same points as used to substantiate the above recommendations, the Discussion section was rewritten at several locations:

On lines 759-768: “However, factors like artemin, NRTN, endothelins, and netrin 1 may be required to recapitulate the target-specific phenotypical sympathetic neuron diversity observed in vivo^111^. For parasympathetic neurons, besides NRTN, TGFβ1, is another promising candidate to improve neuron maturation. in vivo, TGFβ1 regulates the expression of Ca^2+^-activated K^+^-channels together with NRG1^156,157^.”

On lines 786-821: “Besides markers expressed by the mature neurons,autonomic neuron identity was usually supported during differentiation by demonstrating the intermediate presence of NCC markers NGFR, HNK1 and/or SOX10 (or CD49d, which correlates with SOX10 expression^158^). Of these, SOX10 seems most robustly expressed in both human premigratory and migratory human NCCs^87^, but all markers are also expressed in parts of the neural tube, underscoring the need for multiple NCC markers in vitro.

After NCC induction, SCP formation is another specific feature of parasympathetic neuron development. However, only one article demonstrated the intermediate presence of SCPs by SOX10 expression and Schwann cell differentiation potential^47^. Considering that NCCs also express SOX10^87^, markers such as myelin protein 0 and cadherin 19 should be used to distinguish SCPs from NCCs^122^.

Even after establishing intermediate SCP identity, parasympathetic neurons and cholinergic enteric neurons cannot be distinguished from one another. The field currently lacks a comprehensive transcriptomic characterization of parasympathetic neurons^148^, as exists for sympathetic neurons^111^, to address this issue. A recent investigation of the right atrial ganglionic plexus, which contains parasympathetic neurons, may provide some clues^160^. However, the neurons of the intrinsic cardiac ganglia are diverse^161^. Ideally, single-cell RNA sequencing of peripheral cholinergic neurons would reveal specific molecular parasympathetic markers for future PSC-derived parasympathetic protocols.”

On lines 835-857: “From a functional point of view, a major challenge for hPSC-derived neurons is to develop from electrically passive hPSCs to mature neurons within a short time span. Besides prolonged time in culture, other strategies to improve autonomic neuronal maturity include the generation of microtissues containing multiple cell types. As an example, addition of hPSC-derived cardiac fibroblasts to three-dimensional (3D) microtissues consisting of hPSC-CMs and cardiac endothelial cells has been shown to improve hPSC-CM maturity^165^. Co-culture with epicardium-derived cells has also been shown to improve cardiomyocyte maturation and to stimulate sympathetic ganglion neurite outgrowth^10,166^. Satellite glial cells, which support neuronal survival and firing in vivo^167^, are another interesting candidate to assist autonomic neuronal maturation.”

3. The authors completely neglected to address the topic of 3D cell culture vs 2D cell culture. Were all protocols performed in 2D cell cullture? This might be a major limitation to the field, especially when considering the specific anatomy of autonomic neurons (assembled in ganglia with close intimacy of other cells types such as glia and fibroblasts). Please include this in the revised manuscript.

Thank you for this very relevant suggestion. All protocols were indeed developed for 2D culture. We have added a new paragraph to our discussion on lines 858-868:

“Three-dimensional autonomic neuronal models

In addition to glial cells, autonomic neurons neighbor several other cell types in vivo, such as pericytes and endoneurial fibroblasts^168^. Exposure to the signals and extracellular matrix deposited by these cells can influence culture stability, as well as gene and protein expression levels^169^. Ultimately, 3D cultures composed of multiple cell types could potentially model disease phenotypes that two-dimensional (2D) cultures consisting mainly of neurons may not be accurate or sensitive enough for. However, although several articles report the self-assembly of autonomic neurons into ganglia-like clusters^21,30,42^, all current protocols have been developed for 2D culture. Considering the variety of cells that is likely to impact autonomic neuron function, and the 3D anatomy of autonomic ganglia in vivo, the field would benefit from the availability of 3D models of autonomic ganglia with multiple cell types.”

4. Most of the Figures are highly complex and not self-explanatory. While we appreciate the amount of work that went into them and are sure the authors have a well-thought-through colour scheme, please try to make them easier to understand for the general audience. We specifically suggest including (sub)titles, captions and legends for the different colours of bars, arrows and growth factors.

Thank you for the suggestions to increase the impact of our figures. To improve interpretation, we have added titles to all figures, and internal figure legends where relevant. In addition, we have applied more neutral background colours for all figures explaining the in vivo functions of signaling cues. In doing so, we believe that the parallels between the colours used in the protocol overviews and the ‘in vivo*’* figures have become clearer.